# GReaTer: Gradients over Reasoning makes smaller Language Models Strong Prompt Optimizers

**Sarkar Snigdha Sarathi Das**[†]    **Ryo Kamoi**[†]    **Bo Pang**[◇]    **Yusen Zhang**[†]
**Caiming Xiong**[◇]    **Rui Zhang**[†]
[†]The Pennsylvania State University    [◇]Salesforce Research
{sfd5525, rmz5227}@psu.edu

## ABSTRACT

The effectiveness of large language models (LLMs) is closely tied to the design of prompts, making prompt optimization essential for enhancing their performance across a wide range of tasks. Many existing approaches to automating prompt engineering rely exclusively on textual feedback, refining prompts based solely on inference errors identified by large, computationally expensive LLMs. Unfortunately, smaller models struggle to generate high-quality feedback, resulting in complete dependence on large LLM judgment. Moreover, these methods fail to leverage more direct and finer-grained information, such as gradients, due to operating purely in text space. To this end, we introduce GReaTer, a novel prompt optimization technique that directly incorporates *gradient information over task-specific reasoning*. By utilizing task loss gradients, GReaTer enables self-optimization of prompts for open-source, lightweight language models without the need for costly closed-source LLMs. This allows high-performance prompt optimization without dependence on massive LLMs, closing the gap between smaller models and the sophisticated reasoning often needed for prompt refinement. Extensive evaluations across diverse reasoning tasks including BBH, GSM8k, and FOLIO demonstrate that GReaTer consistently outperforms previous state-of-the-art prompt optimization methods, even those reliant on powerful LLMs. Additionally, GReaTer-optimized prompts frequently exhibit better transferability and, in some cases, boost task performance to levels comparable to or surpassing those achieved by larger language models, highlighting the effectiveness of prompt optimization guided by gradients over reasoning. Code of GReaTer is available at: https://github.com/psunlpgroup/GreaTer.

## 1 INTRODUCTION

Large Language Models (LLMs) have demonstrated impressive performance across various task domains (Brown, 2020; Achiam et al., 2023; Reid et al., 2024). However, these models are known to exhibit prompt sensitivity, a phenomenon where slight variations in input prompts can lead to significant differences in output quality (Lu et al., 2021; Madaan & Yazdanbakhsh, 2022; Zhao et al., 2021; Reynolds & McDonell, 2021; Wei et al., 2022; Kojima et al., 2022). Consequently, prompt design has emerged as a critical factor in achieving optimal LLM performance. As the popularity of LLMs has surged, "Prompt Engineering" has become a focal point of attention in the field. Traditionally, this process has been carried out by domain experts who iteratively query expensive LLMs until the desired response is obtained. However, this manual approach is time-consuming and resource-intensive, prompting researchers to explore more efficient alternatives. Recent research has focused on Automated Prompt Engineering (Zhou et al., 2022), which aims to systematically search for prompts that improve target task performance. Following this line of research, (Pryzant et al., 2023; Ye et al., 2023) improved upon it by resorting to computationally expensive stronger LLMs to reason about failure causes in smaller, efficient LLMs deployed in practical tasks. Pryzant et al. (2023) termed this feedback as "textual gradient", since this feedback is leveraged to improve the prompts iteratively.

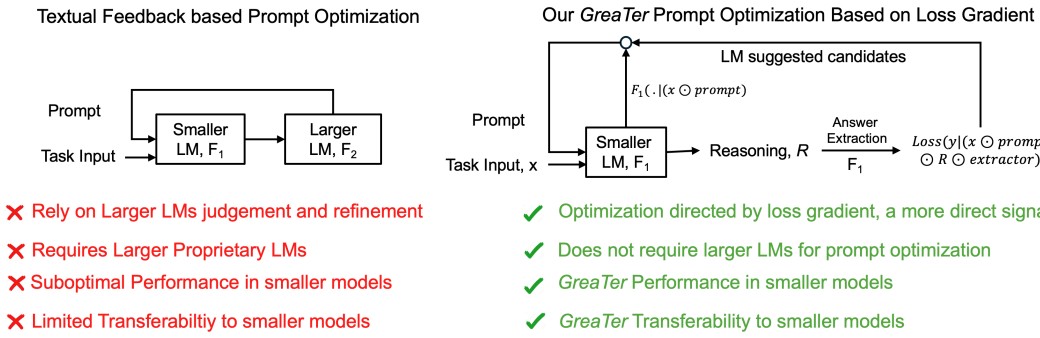

Figure 1: Comparison of textual feedback-based prompt optimization and GREATER. Left: textual feedback relies entirely on a larger language model's judgments. Right: GREATER avoids external large, proprietary models, using token suggestions from a small model and guiding prompt token selection with loss gradients. GREATER incorporates model reasoning by first generating reasoning, then applying an extraction prompt to obtain answer logits for computing loss gradients. This "gradient over reasoning" approach optimizes using direct signals rather than relying on language model feedback.

Despite showing promising performance, the primary limitation of this category of prompt optimization is the reliance on massive LLMs like GPT-4 for optimizing smaller model performance. Smaller LLMs used in practice rely on large models like GPT-4 for optimization, as these big models generate the "textual gradients" needed to refine and transfer knowledge. (Zhang et al., 2024) found that smaller LMs are incapable of generating such optimization feedback, further emphasizing the dependence on large models. Thus, enhancing smaller models depends on the computational power of larger ones. Additionally, the optimization process increases computational costs due to the need for sizeable prompt length due to multiple task samples, and heavy dependence on the optimizer LLM's judgment may result in less reliable outcomes.

To mitigate these issues, we propose GREATER that allows smaller LLMs to optimize prompts using true gradients (i.e., numerical loss gradients) without resorting to larger models. Figure 1 (right) gives an overview of our approach. GREATER leverages *"gradient over reasoning"* for more accurate prompt improvement direction. GREATER first calculates the forward token probabilities to generate a small number of probable token candidates at the selected position conditioned on the input. Then, it utilizes the LLM to generate the reasoning for problem solution, and extracts the final answer logits for calculating the loss. Finally, we leverage the gradient calculated for the probable token candidates to select the best tokens for optimization. Our technique innovates by tackling token discreteness while integrating reasoning chains, crucial for limited datasets like Big Bench Hard (Suzgun et al., 2022), which provide final labels without explicit reasoning paths.

GREATER shows strong prompt optimization performance, where optimized prompt often delivers performance equivalent to larger LLMs in solving the task. In our experiments, we selected `Llama-3-8B-Instruct` (Meta, 2024) and `Gemma-2-9B-it` (Team et al., 2024), two highly popular smaller language models that are proven to be also very useful in solving different tasks. Across a wide variety of selected BBH (Suzgun et al., 2022) tasks, mathematical reasoning task GSM8k, and first-order logic task, FOLIO (Han et al., 2022), GREATER shows up to 8.9% performance improvement over SOTA prompt optimization technique on average in BBH suite of tasks. Moreover, GREATER optimized prompts perform on par or better than GPT-4 optimized prompts (Ye et al., 2023), demonstrating superior performance without resorting to larger proprietary LLMs.

## 2  RELATED WORK

**LLMs as Prompt Optimizers.**   Recently, significant attention has been brought to the prospect of LLMs as prompt optimizers for less powerful models. This line of work was first proposed by Zhou et al. (2022) by prompting LLMs with input-output pairs to infer the instruction. Pryzant et al. (2023) formalized the term "textual gradient" to refer to textual feedback based optimization. Here the authors introduced the use of mini-batches of data to create a natural language feedback. These

gradients critique the current prompt, mimicking the role of numerical gradient in optimization. Later, a large body of works has improved upon it by using optimization logic in text space (Yang et al., 2023; Yuksekgonul et al., 2024), meta-prompt engineering (Ye et al., 2023), agent-based learning and reasoning (Wang et al., 2023; Liu et al., 2024; Shinn et al., 2024), external trained model (Cheng et al., 2023), evolutionary algorithms (Guo et al., 2023; Liu et al., 2024), etc. Hu et al. (2024) introduced ZOPO, leveraging Gaussian processes inspired by the Neural Tangent Kernel to systematically explore local optima in prompt optimization using zeroth-order methods. Other approaches like programming model (Khattab et al., 2023), editing (Prasad et al., 2022), and reinforcement learning (Deng et al., 2022) are also notable. These techniques usually rely on reasoning and judgment from larger LMs to improve the performance of smaller LMs. In other words, the larger model can share its knowledge through the optimized prompt which helps smaller models achieve performance uplift. Therefore, to get strong results with smaller, more lightweight language models, prompts must be optimized using powerful, expensive, closed-source models, as smaller models are inadequate at this optimization on their own (Zhang et al., 2024).

**Prompt Tuning.** Prompt-tuning has been explored in prior works as task-specific continuous vectors tuned by gradient-based methods to improve task performance (Li & Liang, 2021; Lester et al., 2021; Qin & Eisner, 2021; Gao et al., 2020). Discrete prompts on the other hand involve searching for discrete vocabulary tokens through gradients (Shin et al., 2020; Shi et al., 2022). These approaches can be further extended for visual prompt tuning (Wen et al., 2024), where the authors optimize hard text based prompts through efficient gradient-based optimization. A fundamental flaw in these methods stems from the fact that these methods are typically only suitable for classification tasks or tasks with fixed input-output structures, as they rely on predefined templates and verbalizers. Reasoning tasks on the other hand require complex analytical reasoning chains, e.g., Big-Bench-Hard (Suzgun et al., 2022), which leads to the final output, where using a fixed template verbalizer is incompatible and impractical.

**Jailbreaking LLMs.** Gradient-based search methods have also been applied to find trigger prompts that bypass LLM alignment-based filtering and generate harmful responses (Zou et al., 2023). These methods have been further refined to improve readability and effectiveness by including perplexity regularization and constrained-decoding (Guo et al., 2021; Alon & Kamfonas, 2023; Liu et al., 2023; Guo et al., 2024). Similar to prompt tuning, these methods also adhere to a simple input-output structure. The target output typically is an affirmative response *"Sure here is"*, without emphasis on the reasoning chain.

## 3 PROBLEM DEFINITION

We formally define the problem of prompt optimization to lay the foundation of the optimization target. Given a language model $f_{\text{LLM}}$, and a small representative task dataset, $\mathcal{D}_{task} = \{(x_1, y_1), \ldots (x_n, y_n)\}$, the goal of prompt optimization is to find a prompt $p^*$ such that:

$$p^* = \arg\max_p \sum_{(x,y)\in\mathcal{D}_{task}} m\left(f_{\text{LLM}}(x;p), y\right) \tag{1}$$

where $f_{\text{LLM}}(x;p)$ is the output from task language model $f_{\text{LLM}}$ upon channeling the input $x$ with the prompt $p$, and $m(\cdot)$ is the evaluation function for this task.

**Textual Feedback Based Prompt Optimization.** As shown in the left part of Figure 1, to search for $p^*$, previous prompt optimization methods based on textual feedback use an optimizer model $f_{\text{optimizer}}$ which is usually substantially larger and more expensive than $f_{\text{LLM}}$ (Zhou et al., 2022; Ye et al., 2023; Pryzant et al., 2023). Conceptually, $f_{\text{optimizer}}\big(m(f_{\text{LLM}}(x;p), y)|(x, y) \in \mathcal{D}_{task}\big)$ drives the optimization process by assessing and providing feedback for refining the prompt. Therefore, finding $p^*$ primarily relies on the capabilities of $f_{\text{optimizer}}$ and its hypothesis for prompt refinement.

## 4 OUR METHOD

While Section 3 provides the formal explanation of prompt optimization, it's important to understand the role of reasoning in this process. A well-crafted prompt gives clear problem-solving instructions

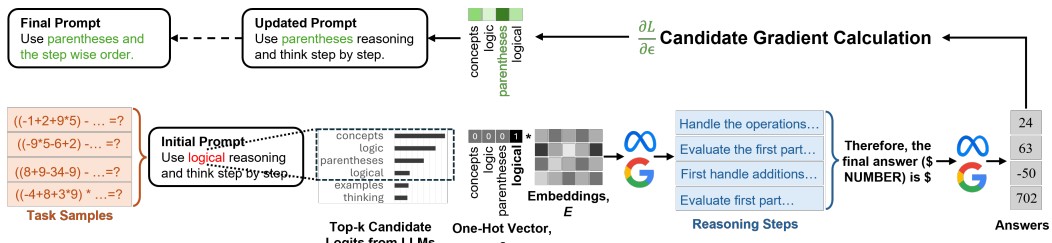

Figure 2: Overall workflow of GREATER. (i) The language model $f_{\text{LLM}}$ generates token candidates by conditioning on input samples. (ii) $f_{\text{LLM}}$ uses task input and current prompt to generate reasoning and extract final answer logits. (iii) The logits are used to calculate loss and compute gradient over generated reasoning with respect to the candidate tokens. These gradients determine the selection of candidate token to update the current position of the current prompt.

that guide the language model to think through the problem in a specific way, helping it arrive at a valid answer. Our method, GREATER, is built on this principle. As shown in Figure 2, GREATER begins by analyzing task examples to propose potential token candidates, essentially exploring different ways to solve the task. The task input and prompts are then used to generate reasoning steps, from which the final answer and loss are extracted. By applying gradients over the generated reasoning, GREATER refines the candidate selection, ensuring it follows the optimal path for improved performance.

## 4.1 METHOD OVERVIEW

Given an input $x$ and a prompt $p = [p_1, p_2, p_3, \ldots]$ consisting of several tokens $p_i$, $f_{\text{LLM}}$ generates the reasoning chain, $r \sim f_{\text{LLM}}(x \odot p)$ for the input ($\odot$ for concatenation). Then, we can extract the final answer from $r$ by prompting $f_{\text{LLM}}$ with a formatted extractor prompt $p_{\text{extract}}$, e.g., *"Therefore, the final answer ($\$NUMBER\$$) is $\$$"*. Consequently, this produces the final answer logits, $y' = f_{\text{LLM}}(x \odot p \odot r \odot p_{\text{extract}})$. We define our loss function as:

$$\mathcal{L} = \mathcal{L}_{CE}\left(f_{\text{LLM}}\left(x \odot p \odot r \odot p_{\text{extract}}\right), y\right) \tag{2}$$

In Eq. 2, for a fixed $p_{\text{extract}}$ and $x$, only $p$ will affect the loss; therefore we can calculate $\frac{\partial \mathcal{L}}{\partial p}\big|_{x, p_{\text{extract}}}$. This loss gradient takes the reasoning into account, making it a more direct signal to drive the prompt optimization process. Therefore, this "gradient over reasoning" can be a highly potent alternative to current textual feedback based prompt optimization that entirely relies on massive LLM feedback. Note that, in Eq. 2, the entire chain is equivalent to one single forward pass through $f_{\text{LLM}}$, since key, values from reasoning generation can be cached and utilized during logit extraction.

We sequentially apply GREATER optimization in each token position $p_i$ of $p$. GREATER first employs a *Candidate Proposal* (Section 4.2) stage, where it uses language model-guided forward probabilities to generate potential token candidates for optimized prompt. Subsequently, the current prompt is channeled through LLM, $f_{\text{LLM}}$ for solution generation, and then answer logits extraction (Section 4.3). Using these logits, we can calculate the loss and gradient with respect to the small number of prompt token candidates which we use for token selection. (Section 4.4). This process can be applied over all the prompt token positions repeatedly.

In the following subsections, we discuss each stage of GREATER to optimize prompt token $p_i$ at position $i$. Algorithm 1 shows all the steps in GREATER. This process is sequentially applied over all the positions repeatedly until convergence.

## 4.2 PROMPT TOKEN CANDIDATE PROPOSAL

For optimizing prompt token $p_i$, we first leverage the task language model (LM) $f_{\text{LLM}}$ probabilities to propose candidates for the position. For a sample input $x_j \in \mathcal{D}_{task}$, we can calculate the top-$k$ probabilities for candidate token proposals:

$$\text{cand}_{i,j} = \text{top-}k\left[f_{\text{LLM}}(\cdot | x_j \odot p_1, p_2, \ldots, p_{i-1})\right] \tag{3}$$

---

**Algorithm 1** GREATER

1: **Input:** Initial Prompt $p_{\text{init}} = (p_1, p_2, p_3, \dots)$, Task Dataset $\mathcal{D}_{\text{task}} = (X, Y)$, Language Model $f_{\text{LLM}}$, Extractor Prompt $p_{\text{extract}}$
2: **Output:** Optimized Prompt $p^*$
3: $p, p^*, L_{\text{best}} \leftarrow p_{\text{init}}, p_{\text{init}}, \infty$
4: position, $i \leftarrow 0$
5: **for** $t \leftarrow 1$ to $T$ **do**
6:     *// Candidate Proposal Stage*
7:     candidates$_i \leftarrow \bigcap_{x_j \in \mathcal{D}_q}$ cand$_{i,j}$, where $\mathcal{D}_q \subset \mathcal{D}_{\text{task}}$        $\triangleright$ From Eqn. 3 and Eqn. 4
8:     Compute one-hot token indicator $\epsilon_i$ for candidates$_i$        $\triangleright$ From Section 4.2
9:     *// Reasoning Generation and Extraction*
10:    Calculate logits, $\hat{Y} \leftarrow \{ f_{\text{LLM}}(x \odot p \odot f_{\text{LLM}}(x \odot p) \odot p_{\text{extract}}) | x \in \mathcal{D}\}$    $\triangleright$ Extract Solution and Logits (From Eqn. 5)
11:    *// Gradient Over Reasoning driven Candidate Selection*
12:    Compute total Loss $\mathcal{L}(\hat{Y}, Y)$ for $\mathcal{D}$ and $\frac{\partial \mathcal{L}}{\partial \epsilon_i}$        $\triangleright$ From Eqn. 7
13:    *// select top-$\mu$ tokens with gradients and find the candidate with the lowest loss*
14:    selections$_i \leftarrow$ token[top-$\mu(-\frac{\partial \mathcal{L}}{\partial \epsilon_i})$], where $\mu = 3$
15:    $p_i, \mathcal{L}_{min} \leftarrow \arg\min_{c \in \text{selections}_i} \sum_{(x,y) \in \mathcal{D}} \mathcal{L}\left(f_{\text{LLM}}(x \odot p_{<i} \odot c \odot p_{>i}), y\right)$    $\triangleright$ Find the best candidate, assuming $\arg\min$ also returns min_loss
16:    **if** $\mathcal{L}_{min} < L_{\text{best}}$ **then**
17:       $L_{\text{best}} \leftarrow \mathcal{L}_{min}$        $\triangleright$ Update the best loss
18:       $p^* \leftarrow p$        $\triangleright$ Update the best prompt
19:    $i \leftarrow (i + 1) \mod \text{length}(p_{\text{init}})$
20: **return** $p^*$

---

Therefore, we take top-$k$ tokens from $f_{\text{LLM}}$ conditioning on $x_j$ followed by the previous tokens from the optimized prompt. We calculate cand$_{i,j}$ for a randomly sampled set of $q$ inputs, $\mathcal{D}_q \subset \mathcal{D}_{\text{task}}$. Since we want the token candidates to be relevant for all samples in the task, token candidates for position $i$ will incorporate candidates for all samples, so

$$\text{candidates}_i = \bigcap_{x_j \in \mathcal{D}_q} \text{cand}_{i,j} \tag{4}$$

Equation 4 gives candidates$_i$, a set of promising token candidates for position $i$. Since candidates$_i$ are suggested by LM and conditioned on the inputs, they are representative of the problem domain and more interpretable. Therefore, gradient mass will be calculated only for a small number of promising token candidates, $\epsilon_i$, preventing it from being dispersed over the entire vocabulary.

**One-Hot Token Indicators.** As shown in Figure 2 (i), a one-hot token indicator $\epsilon_i$ is created for only candidates$_i$ and the current token $p_i$, with a value of one only for $p_i$ and zeros for all other candidates. Concurrently, from the original LM embedding table, we extract only the rows that correspond to $\epsilon_i$ to create a subset of the original embedding table, $E$. This table is then multiplied by the one-hot indicator $\epsilon_i$, allowing the input to be passed through $f_{\text{LLM}}$.

## 4.3 REASONING GENERATION AND EXTRACTION

A straightforward way of calculating loss and subsequently taking gradient over it is to only consider the output of $f_{\text{LLM}}(x \odot p)$ and taking cross-entropy loss of the output token to that of the ground truth. However, this completely ignores the role of reasoning as discussed before, whereas modern language models require the generation of a complex reasoning chain to generate correct output (Wei et al., 2022). Therefore, simply considering $f_{\text{LLM}}(y|x \odot p)$ would give us the wrong objective to optimize, which in turn will give incorrect gradient information.

Consequently, in this stage, we first generate a reasoning $r \sim f_{\text{LLM}}(x \odot p)$. In $r$, the language model generates reasoning to derive the final answer. To compute the loss accurately, we must extract the answer logits and compare with the ground truth label. A simple way to do that is to get $f_{\text{LLM}}$ logits when conditioned on the input, reasoning, followed by a formatted extractor prompt, $p_{\text{extract}}$.

$$\hat{y} = f_{\text{LLM}}(x \odot p \odot r \odot p_{\text{extract}}) \tag{5}$$

Equation 5 results in LM logits that consider the reasoning generated from the current prompt $p$. Therefore, it better represents the reasoning chain making it suitable for loss calculation.

## 4.4 GRADIENT OVER REASONING DRIVEN CANDIDATE SELECTION

Equation 5 gives us the logits $\hat{y}$ that incorporate reasoning chain information originating from current $p$. Therefore, we can optimize the cross-entropy loss as below. Additionally, we add the perplexity regularization term $\mathcal{L}_{perpl}$ to promote the interpretability of the optimized prompt similar to (Zou et al., 2023; Guo et al., 2024).

$$\mathcal{L}_{\text{CE}} = \text{cross\_entropy}(\hat{y}, y), \mathcal{L}_{\text{perpl}} = \exp\left(-\frac{1}{|p|}\sum_{i=1}^{|p|}\log f_{\text{LLM}}(p_i \mid x, p_{<i})\right), \mathcal{L} = \mathcal{L}_{\text{CE}} + \lambda\mathcal{L}_{\text{perpl}} \tag{6}$$

However, the perplexity term is less important in our case given that we are optimizing only over the top-$k$ candidates suggested by the LM. Consequently, the candidate proposal stage handles most of the interpretability duties.

Upon calculating the loss $\mathcal{L}$, we can do a backward pass to calculate the gradient over the generated reasoning with respect to one-hot token indicator $\epsilon_i$. As $\frac{\partial\mathcal{L}}{\partial\epsilon_i}$ gives us the gradient directions for each of the token candidates in Equation 4, the token replacement at position $i$ is given by:

$$p_i = \text{token}[\arg\max_{\epsilon_i}(-\frac{\partial\mathcal{L}}{\partial\epsilon_i})] \tag{7}$$

In practice, we select the top-three candidate tokens with the highest negative gradients and evaluate the token replacement with a forward pass on the training set. This ensures the robustness of the replacement selection process leading to higher performance from optimization.

## 5 EXPERIMENTS

In this section, we demonstrate that GREATER is highly effective in prompt optimization delivering substantial performance improvement across different tasks. Section 5.1 describes the experiment setup. Section 5.2 presents the main results of the GREATER performance with smaller language models. In Section 5.3, we compare GREATER prompts optimized by smaller language models against the prompts optimized by larger proprietary language models using state-of-the-art baseline methods. Section 5.4 performs an ablation study on the effectiveness of gradient over reasoning in GREATER. Section 5.5 demonstrates the transferability of GREATER prompts. Section 5.6 shows some case studies.

## 5.1 EXPERIMENT SETUP

**Datasets.** To evaluate the efficacy of our approach, we use **GSM8K** (Cobbe et al., 2021), **Big-Bench-Hard (BBH)** (Suzgun et al., 2022), and **FOLIO** (Han et al., 2022) benchmark datasets for diverse reasoning tasks in mathematics, commonsense, and logical reasoning. For GSM8K, we used the same test split as (Cobbe et al., 2021). For BBH, we used the similar split setting as (Yuksekgonul et al., 2024). For FOLIO, we evaluate the natural language reasoning with the first-order logic task for the evaluation which also requires complex reasoning capabilities. We used the natural language reasoning task with premises text, conclusion text to infer the labels, and we use the validation set of the updated version of FOLIO[1]. More details about these are given in Appendix A.1.

**Models.** To demonstrate that our approach can generalize to different backbone language models, we choose two strong and widely-used open-source language models: **Llama-3-8B** (Meta, 2024) and **Gemma-2-9B** (Team et al., 2024). For both models, we use the instruction tuned versions [2] [3].

---

[1]https://huggingface.co/datasets/yale-nlp/FOLIO

[2]https://huggingface.co/meta-llama/Meta-Llama-3-8B-Instruct

[3]https://huggingface.co/google/gemma-2-9b-it

Table 1: Overall results. GREATER brings substantial performance improvements across different reasoning tasks, demonstrating its efficacy in prompt optimization with smaller models. It considerably outperforms state-of-the-art prompt optimization methods. Detailed prompts and results with breakdown across all the tasks are shown in Appendix H and Appendix I.

| Method | Gemma-2-9B | | | Llama-3-8B | | |
|---|---|---|---|---|---|---|
| | GSM8K | BBH | FOLIO | GSM8K | BBH | FOLIO |
| ZS-CoT (Kojima et al., 2022) | 88.6 | 71.7 | 65.0 | 79.6 | 62.2 | 58.6 |
| APE (Zhou et al., 2022) | 88.6 | 71.7 | 67.5 | 79.9 | 63.1 | 57.6 |
| APO (Pryzant et al., 2023) | 88.6 | 72.3 | 63.1 | 81.1 | 62.7 | 58.6 |
| PE2 (Ye et al., 2023) | 88.6 | 68.9 | 62.1 | 80.1 | 61.5 | **62.6** |
| TextGrad (Yuksekgonul et al., 2024) | 87.8 | 72.9 | 67.5 | 78.5 | 58.5 | 56.2 |
| GREATER | **89.4** | **76.6** | **69.1** | **82.6** | **68.7** | **62.6** |

**Baselines.** We compare with state-of-the-art prompt optimization baselines: **APE** (Zhou et al., 2022), **APO** (Pryzant et al., 2023), **PE2** (Ye et al., 2023), **TextGrad** (Yuksekgonul et al., 2024) where we use them on Llama-3-8B and Gemma-2-9B for optimization. In addition, we also show the efficacy of our approach by comparing against the prompts optimized by massive proprietary LLMs (e.g., GPT-4, PaLM-2-L) using APE, APO, PE2, TextGrad, **OPRO** (Yang et al., 2023), **Evo-Prompt** (Guo et al., 2023).

## 5.2 OVERALL RESULTS

We evaluate the performance of GREATER in comparison to state-of-the-art baselines, including APE, APO, PE2, TextGrad, and the original zero-shot chain-of-thought approach (Kojima et al., 2022), across the GSM8K, BBH, and FOLIO benchmarks. Table 1 highlights the significant and consistent improvements achieved by GREATER over state-of-the-art prompt optimization methods, especially when optimizing prompts using lightweight language models. Across both Gemma-2-9B and Llama-3-8B, GREATER demonstrates remarkable stability and outperforms the baselines. This is particularly noteworthy given the high variability in performance seen in these textual feedback methods, which tend to be unreliable with smaller models due to increased prompt sensitivity. For instance, while TextGrad performs reasonably well on BBH with Gemma-2-9B, it falters significantly with Llama-3-8B. In contrast, GREATER delivers outstanding results across the board, excelling across diverse tasks. Note that, in the FOLIO dataset, GREATER performs on par with PE2 (Ye et al., 2023), suggesting that the model has likely reached its performance ceiling, leaving minimal room for further improvement.

## 5.3 COMPARISON WITH PROMPTS OPTIMIZED BY LARGER PROPRIETARY MODELS

Besides optimization performance in lightweight language models, we also evaluate the efficacy of GREATER optimized prompt compared with the prompts optimized by massive LLMs. These massive models, like GPT-4 and PaLM-2-L, inherently possess deeper knowledge of complex reasoning tasks and often provide richer guidance in their optimized prompts, making them strong baselines to compare with. In Table 2, we compare the performance of GREATER optimized prompt to that of APE (optimized by GPT-4), APO (optimized by GPT-4), PE2 (optimized by GPT-4), OPRO (optimized by PaLM-2-L), EvoPrompt(optimized by GPT-3.5). We use GSM8K, and five randomly selected tasks from BBH to compare the performance of these prompts. We see that in both Llama-3-8B and Gemma-2-9B, GREATER optimized prompts perform substantially better and robustly compared to the baseline prompts optimized with massive LLMs. The consistency in performance uplift makes GREATER a viable choice to boost task performance in reasoning tasks with lightweight language models.

## 5.4 ABLATION OF GRADIENT OVER REASONING

As outlined in Section 4.3 and Section 4.4, a core feature of GREATER is the concept of *Gradient Over Reasoning*. Conventional gradient-based optimization methods rely solely on the target answer labels to compute the loss and gradients, overlooking the crucial role that reasoning

Table 2: Comparison of GREATER with prompts optimized by larger proprietary LLMs. GREATER performs on par with or notably better than prompts optimized by GPT 4 and PaLM-2-L across GSM8K and five randomly chosen BBH tasks using Llama-3-8B and Gemma-2-9B. EvoPrompt does not report its prompts on GSM8K. Here, *Target Model: Llama-3-8B* and *Method (Optimized by): APE (GPT-4)* indicates that Llama-3-8B was used for prompt evaluation while the prompt was optimized by GPT-4 with APE.

| Target Model | Method (Optimized by) | GSM8K | BBH (5 randomly chosen tasks) | | | | | |
|---|---|---|---|---|---|---|---|---|
| | | | movie_rec. | object_count. | tracking_five. | hyperbaton | causal | Average |
| Llama-3-8B | APE (GPT-4) | 80.7 | 50 | 82 | 50 | 76 | 56 | 62.8 |
| | EvoPrompt (GPT-3.5) | - | 48 | 74 | 42 | 68 | 48 | 56.0 |
| | APO (GPT-4) | 81.1 | 56 | 68 | 49 | 75 | 51 | 59.8 |
| | PE2 (GPT-4) | 81.5 | 48 | 82 | 45 | 79 | 49 | 60.6 |
| | OPRO (PaLM-2-L) | 82.3 | 60 | 78 | 40 | 70 | **57** | 61.0 |
| | GREATER (Llama-3-8B) | **82.6** | **57** | **90** | **70** | **84** | 57 | **71.6** |
| Gemma-2-9B | APE (GPT-4) | 89.2 | 48 | 61 | 83 | 83 | 60 | 67.0 |
| | EvoPrompt (GPT-3.5) | - | 51 | 70 | 82 | 83 | 61 | 69.4 |
| | APO (GPT-4) | 89.3 | 52 | 84 | 72 | 82 | 59 | 69.8 |
| | PE2 (GPT-4) | **89.6** | 50 | 65 | 71 | 84 | **64** | 66.8 |
| | OPRO (PaLM-2-L) | 89.0 | 50 | 58 | 76 | 81 | 58 | 64.6 |
| | GREATER (Gemma2-9B) | 89.4 | **56** | **87** | **85** | **88** | 61 | **75.4** |

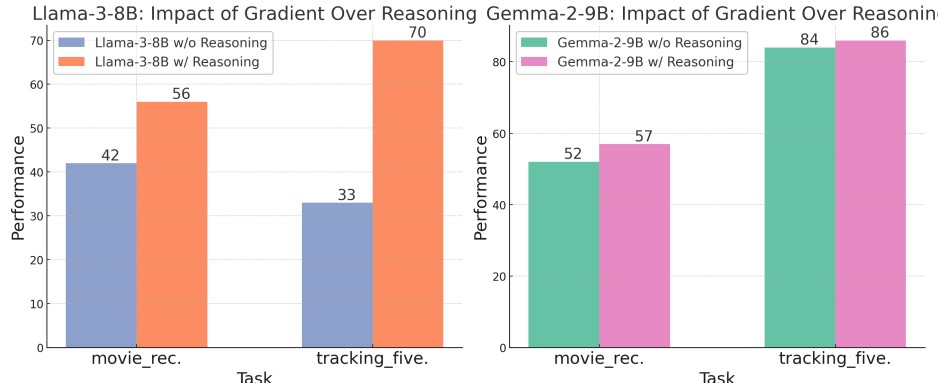

Figure 3: Ablation study on "Gradient Over Reasoning" in GREATER. Gradient calculation without reasoning causes notable performance drops, showing the importance of reasoning for gradients.

plays in this process. To highlight the significance of incorporating reasoning into gradient calculations, we conducted a comparative analysis of GREATER with and without applying *Gradient Over Reasoning*, while keeping all other steps identical. For this experiment, we optimized with both Llama-3-8B and Gemma-2-9B on two BBH tasks: `movie_recommendation` and `tracking_shuffled_objects_five_objects` task. Figure 3 shows the comparison between with and without *gradient over reasoning*. It clearly demonstrates a substantial performance drop when *Gradient Over Reasoning* is omitted. Fundamentally, from Equation 5, removing the reasoning generation part $r$ equates to calculating an incorrect objective function, which will drive gradient-based token selection to diverge from the optimal solution.

## 5.5 PROMPT TRANSFERABILITY

We conduct further experiments to evaluate the transferability of GREATER optimized prompt. In Section 5.5.1, we first evaluate the transferability of GREATER prompt between two smaller language models, Llama-3-8B and Gemma-2-9B, by evaluating the performance of Gemma-2-9B optimized prompts on Llama-3-8B, and vice-versa. Next, in Section 5.5.2, we evaluate the transferability of prompts from a smaller model (Llama-3-8B) to a much larger model (Gemma-2-27B).

Table 3: Transferability of Llama-3-8B optimized prompts to Gemma-2-9B (Upper) and vice versa (Lower). The results demonstrate that prompts produced by GREATER exhibit strong transferability compared with those produced by other state-of-the-art prompt optimization methods.

| Target Model | Method (Optimized by) | BBH (5 randomly chosen tasks) | | | | | |
|---|---|---|---|---|---|---|---|
| | | movie_rec. | object_count. | tracking_five. | hyperbaton | causal_judgement | Average |
| | | Llama-3-8B → Gemma-2-9B | | | | | |
| Gemma-2-9B | TextGrad (Llama-3) | 53 | 78 | 56 | 84 | **63** | 66.8 |
| | APO (Llama-3) | 53 | 84 | 68 | 84 | 58 | 69.4 |
| | PE2 (Llama-3) | 54 | 84 | 68 | 82 | 60 | 69.6 |
| | GREATER (Llama-3) | **55** | **90** | **85** | **91** | 60 | **76.2** |
| | APO (GPT-4) | 52 | 84 | 72 | 82 | 59 | 69.8 |
| | | Gemma-2-9B → Llama-3-8B | | | | | |
| Llama-3-8B | TextGrad (Gemma-2) | 35 | 29 | 49 | 65 | 36 | 42.8 |
| | APO (Gemma-2) | 54 | 69 | 48 | 71 | **53** | 59.0 |
| | PE2 (Gemma-2) | 56 | 69 | 49 | 50 | **53** | 55.4 |
| | GREATER (Gemma-2) | **58** | **87** | **56** | 70 | 52 | **64.6** |
| | APO (GPT-4) | 56 | 68 | 49 | **75** | 51 | 59.8 |

Table 4: Transferability of Llama-3-8B optimized prompts to Gemma-2-27B. The results demonstrate that GREATER optimized prompts exhibit strong transferability from smaller to larger language models.

| Target Model | Method (Optimized by) | BBH (5 randomly chosen tasks) | | | | | |
|---|---|---|---|---|---|---|---|
| | | movie_rec. | object_count. | tracking_five. | hyperbaton | causal_judgement | Average |
| | | Llama-3-8B → Gemma-2-27B | | | | | |
| Gemma-2-27B | PE2 (Llama-3) | 59 | **92** | 83 | 73 | 54 | 72.2 |
| | APO (Llama-3) | 53 | **92** | 83 | 72 | **58** | 71.6 |
| | Ours (Llama-3) | 59 | 91 | **86** | 82 | 57 | **75.0** |
| | APO (GPT-4) | **64** | **92** | 81 | 77 | **58** | 74.4 |

### 5.5.1 TRANSFER BETWEEN GEMMA-2 AND LLAMA-3

As shown in Table 3, GREATER exhibits exceptional transferability across smaller models compared to other methods. Notably, prompts optimized by GREATER significantly outperform even APO prompts, which were optimized using GPT-4, as reported by Pryzant et al. (2023). This distinction is particularly important, as it highlights that GREATER optimized prompts are not only more effective but also better suited for broader, more generalized use across smaller models. These results clearly emphasize the efficacy and versatility of our approach.

### 5.5.2 TRANSFER TO LARGER MODELS

To assess the transferability of GREATER-optimized prompts to much larger models, we evaluated their efficacy using the Gemma-2-27B model. As demonstrated in Table 4, the GREATER-optimized prompts continue to exhibit strong performance when compared to baseline methods. However, a notable observation is that the performance gap between GREATER and the baselines narrows as the model size increases. Additionally, we notice that APO GPT-4 optimized prompts yield similar performance with GREATER across various tasks, often resulting in closely matched outcomes.

Tables 3 and 4 reveal that optimized prompts enable smaller models to achieve performance comparable to larger ones, highlighting an efficient approach to maximize model capabilities without relying on resource-intensive models.

### 5.6 CASE STUDY

One of the key characteristics of GREATER is that the generated prompts tend to be more varied and innovative compared to the textual feedback-based SOTA baselines. As noted in Table 11 and Table 14, the optimized prompts from traditional approaches are often verbose variations of the original query or standard Chain of Thought (CoT) prompts. In contrast, GREATER prompts frequently

Table 5: Example prompts (abridged) generated by GREATER and APO. GREATER prompts guide structured ways to solve tasks, leading to improved task performance compared to traditional Chain of Thought (CoT) prompts and their variations often generated by textual feedback-based optimization methods like APO. More examples can be found in the Appendix H and I.

| Task | Optimized Prompt by GREATER | Optimized Prompt by APO |
|---|---|---|
| llama3-formal_fallacies | Use formal notation and and think step . . . | Analyze the argument step by step considering premises, logical . . . |
| llama3-causal_judgement | Use causal diagram... | Analyze the situation by identifying the direct and indirect causes . . . |
| llama3-object_counting | Use only addition. Add think step by ... | Let's think step by step. |
| llama3-navigate | Use your reasoning here. I would like numbers assigned.. to.. To represent moving | Analyze the instructions step by step, considering each action's ... |
| llama3-sports_understanding | Use the context or a sentence similar prior knowledge. Assume you a journalist, I would have been covering NHL hockey in Min- nesota before joining this assignment to report sports. | Assess the plausibility of the sentence, considering both literal and figurative meanings, as well as context and domain knowledge. Evaluate the sentence's coherence and relevance to the given context . . . |
| gemma-multistep_arithmetic_two | Use parentheses and and the step wise order... | Let's think step by step. |
| gemma-date_understanding | Use your format Excel formula for this answer to find it... | Let's think step by step. |
| gemma_reasoning_colored | Use your logic. Please answer. person . . . | Analyze the given text and answer . . . |

provide highly insightful guidelines that better aid in problem-solving. Table 5 demonstrates a selection of abridged prompts generated by GREATER and APO. For tasks like formal_fallacies and causal_understanding, GREATER prompts encourage logical analysis rather than mere language-based reasoning, which enhances performance. Similarly, in the object_counting and navigate tasks, the prompts simplify the original problem by converting it into a more straightforward mathematical task, leading to significant improvements. More interestingly, in the sports_understanding task, the prompt promotes agentic behavior, encouraging the model to take an active role in solving the task. Additionally, for multistep_arithmetic_two, the prompt focuses on parentheses to avoid errors in computation, in date_understanding task it instructs to convert into a programming problem (by using Excel formula), and in reasoning_about_colored_objects focuses on logical analysis. These prompts are comparatively more interesting than prompts generated by textual feedback (e.g. APO) where most prompts only show verbose instruction regarding the original question, without any guidance of how to solve the problem. They also demonstrate that GREATER prompts do more than just provide basic instructions—they offer clear and practical strategies tailored to each task. By helping the model think about problems in a more structured and systematic way, GREATER leads to better problem-solving and improved performance compared to traditional methods.

While GREATER generates highly effective and innovative prompts, they may occasionally exhibit minor grammatical issues or a more informal tone compared to standard prompts. For instance, the navigate task and multistep_arithmetic_two task in Table 5 reflect such characteristics. However, these issues can be mitigated by adjusting the Top-$k$ parameter in Equation 3. Incorporating dynamic Top-$k$ selection could further enhance the naturalness and accuracy of the prompts.

## 6 CONCLUSION

We present GREATER, a novel gradient-guided prompt optimization technique that enhances performance without relying on massive proprietary LLMs. GREATER proposes token candidates using sample inputs, generates reasoning, and extracts final answer logits to compute loss and gradients. This "Gradient over Reasoning" provides a strong signal for selecting optimal candidates, enabling significant performance gains on tasks like BBH, GSM8K, and FOLIO, outperforming state-of-the-art textual feedback-based methods for smaller models. Additionally, GREATER prompts demonstrate notable transferability across smaller models and often match the performance of larger models. Future directions include integrating GREATER with textual feedback-based methods for more robust and effective prompt optimization.

ACKNOWLEDGMENT

This work is supported in part by NSF grant 2338418.

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

## A    Benchmark Datasets, Models, and Baselines

In this section, we discuss the details of Datasets, Models, and Baselines.

### A.1    Datasets

As discussed in Section 5.1, we use GSM8K (Cobbe et al., 2021), BBH (Suzgun et al., 2022), and FOLIO (Han et al., 2022) datasets. For GSM8K, we used 100/100 for train/dev set, and original test set of 1319 size. Then, for BBH datasets, we used 21 selected BBH tasks as in Table 11 and Table 14. This covers almost all types of tasks in BBH dataset. We skip word_sorting and dyck_languages tasks from our evaluation since we found that the smaller LLM outputs are very difficult to reliably evaluate due to highly inconsistent output pattern. Finally, for logical_deduction and tracking_shuffled_objects, we select five objects tasks as representative of the tasks. For all tasks we use 50/100/100 train/dev/test splits similar to (Yuksekgonul et al., 2024).

Finally for the FOLIO dataset, we used the latest version of FOLIO (Han et al., 2022) for our evaluation. We use the natural language reasoning task with premises text, conclusion text to infer the labels. The original validation split (203 rows) are used for the evaluation of FOLIO, whereas 50/100 samples are taken for train and dev set respectively out of the original train split.

### A.2    Models

As described in experimentation setup, we only used Llama-3-8B-Instruct and Gemma-2-9B-it in our experiments due to their smaller footprint and strong performance. However, in several tasks Gemma-2-9B-it shows markedly stronger performance than Llama-3-8B, although the memory requirements for Gemma-2-9B-it is substantially higher. As we have shown, stronger models get lesser benefit from prompt optimization, as good prompts only offset for lower capabilities of weaker models. Across our experiments, we also notice substantial improvement in Llama-3-8B-Instruct, whereas the margin gets narrower for the stronger Gemma-2-9B-it.

### A.3    Baselines

Our primary baselines are APO (Pryzant et al., 2023), iterative APE (Kojima et al., 2022), PE2 (Ye et al., 2023), TextGrad (Yuksekgonul et al., 2024). For APO, iterative APE, and PE2 we use the

original implementation by (Ye et al., 2023) for benchmarking. And we used TextGrad's own library for evaluating their results. Other than that, we also compared against original Zero-Shot CoT (Wei et al., 2022), and the larger model optimized reported prompts from APE, APO, PE2, OPRO (Yang et al., 2023), EvoPrompt (Guo et al., 2023).

## B    ADDITIONAL EXPERIMENTATION DETAILS

As shown in Algorithm 1, we run GREATER for $T = 105$ steps with $k = 10$ for top-$k$, $q = 5$ and $\lambda = 0.2$ (in Eq. 6). While Algorithm 1 shows fixed length prompt optimization, we allow dynamic prompt. To attain that, we look for the presence of ending token in the last position. If no ending token is found, we continue the optimization process by adding a placeholder token and optimizing over it. As ending token is encountered we move over to the first position for optimization. We run all our experiments on 2X NVIDIA A100 80GB GPUs.

## C    PERFORMANCE COMPARISON WITH OTHER GRADIENT-BASED METHODS

GREATER leverages the Gradient Over Reasoning chain to identify the optimal prompt required for enhancing task performance. Previous works, such as Shin et al. (2020), utilized gradient-guided search to discover trigger tokens for improving performance in relatively straightforward tasks like sentiment analysis and natural language inference. This concept has also been extended to text-to-image tasks by incorporating gradient-based learning in embedding spaces (Wen et al., 2024). However, these approaches are limited to simpler tasks and fail to address tasks requiring complex reasoning. As demonstrated in Table 6, GREATER-optimized prompts deliver a substantial performance boost compared to prior methods that do not incorporate the reasoning chain.

Table 6:    Comparison of performance in movie_recommendation and tracking_shuffled_objects_five_objects for prompts optimized on Llama-3-8B and Gemma-2-9B. The results demonstrate that prompts optimized by GREATER outperform other methods across both models.

| Method | Llama-3-8B | | Gemma-2-9B | |
| --- | --- | --- | --- | --- |
| | movie_rec. | tracking_five | movie_rec. | tracking_five |
| AutoPrompt (Shin et al., 2020) | 51 | 38 | 50 | 73 |
| PEZ (Wen et al., 2024) | 42 | 35 | 45 | 73 |
| GREATER | **56** | **70** | **57** | **86** |

## D    COMPLEXITY COMPARISON: GREATER VS. TEXT-BASED FEEDBACK APPROACHES

Assume each sample contains $L$ input-output tokens on average.

FOR GREATER

- **Operations Per Sample**:
    - Forward Pass: $O(L^2)$
    - Backward Pass: $O(L^2)$
- **Total Complexity**:
    - For $N$ samples, one forward pass, and one backward pass are performed per sample in a single iteration.
    - Total complexity:
$$N \cdot O(L^2) = O(NL^2)$$

FOR TEXT-BASED FEEDBACK APPROACHES

- **Overview**:
    - No backpropagation stage.
    - Each iteration involves, evaluating all samples, identifying incorrect samples ($n$ incorrect samples), and chaining incorrect samples into a single sequence to generate feedback.

- **Complexity Breakdown**:
    - **First Forward Pass**:
    $$O(NL^2)$$
    - **Feedback Generation**:
        * Incorrect samples ($n$) are chained together.
        * Chained length: $n \cdot L$.
        * Feedback generation complexity:
        $$O((nL)^2) = O(n^2L^2)$$

- **Total Complexity**:
$$O(NL^2) + O(n^2L^2)$$

- **Behavior Based on Task Difficulty**:
    - **Simple Tasks** ($n \to 0$):
    $$\text{Total Complexity} \to O(NL^2)$$
    - **Difficult Tasks** ($n \to N$):
    $$\text{Total Complexity} \to O(NL^2) + O(N^2L^2) = O(N^2L^2)$$

COMPARISON

- **GreaTer Complexity**: $O(NL^2)$ (consistent).
- **Text-Based Feedback Complexity**: Can scale up to $O(N^2L^2)$.

This complexity difference is also translated into real-world performance. In $movie\_recommendation$ task, GREATER is required to be optimized for ∼5 hours, whereas TextGrad (Yuksekgonul et al., 2024) required a total of ∼14 hours for prompt optimization in our own setup.

## E PROMPT OPTIMIZATION VS. FEW-SHOT IN-CONTEXT LEARNING

In-context learning has proven to be highly effective for reasoning tasks in large language models. This raises the question of whether prompt optimization provides any advantages over in-context learning for smaller models. In Figure 4, we compare the performance of five-shot in-context learning with zero-shot reasoning using an optimized prompt from GREATER in Llama-3-8B-Instruct. The results demonstrate that GREATER offers a significant performance improvement over five-shot reasoning. Moreover, using an optimized prompt eliminates the need for repeated input-output examples during inference, leading to greater efficiency.

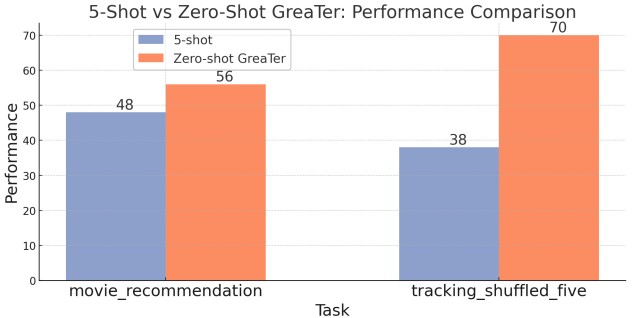

Figure 4: Efficacy of GREATER in zero-shot setting compared to five-shot inference with Llama-3-8B-Instruct.

Table 7: Impact of Initialization Prompt. We can see that different initialization has resulted in different optimized prompt, however they offer comparable performance.

| Initialization Prompt | Optimized Prompt | Score |
|---|---|---|
| Default (*Use proper logical ...*) | Use movie ratings data available here above movies for reference. This HOFF has an interesting analysis based solely on options to options based on movies ratings. Expect from the other movies you are asked, choose option from those mentioned below. | 56 |
| Misleading (Use no thinking just feeling.) | Use one one-liner and explain stepwise for why. ONLY READING IS ALLOWABLE AND NO CHATTY CHAT OR EXCL. | 55 |

Table 8: Performance comparison across selected BBH tasks for Llama-3.2-1B (target model for evaluation) with prompts optimized by different methods. The results demonstrate that prompts optimized by GRᴇᴀTᴇʀ outperform other optimized prompts.

| Method (Optimized By) | movie_rec. | causal_judgement | hyperbaton | tracking_five | object_count. | AVG |
|---|---|---|---|---|---|---|
| APO (GPT4) | 23 | 45 | 61 | 24 | 51 | 40.8 |
| PE2 (GPT4) | 27 | 54 | **69** | 21 | 59 | 46.0 |
| iAPE (GPT4) | 32 | 57 | 53 | 15 | 41 | 39.6 |
| GRᴇᴀTᴇʀ (Llama-3.2-1B) | **46** | **69** | 62 | **24** | **67** | **53.6** |

## F   EFFECT OF INITIALIZATION

For GRᴇᴀTᴇʀ, we start with a fixed prompt: "Use proper logical reasoning and think step by step. Finally, give the actual correct answer." This fixed initialization is convenient for adapting to various tasks. However, we also explore whether using a different initialization affects performance. To test this, we conduct a small-scale experiment with Llama-3-8B on the `BBH-movie-recommendation` task. Table 7 highlights the impact of different prompt initializations on optimization. As shown, a completely misleading initialization leads to a vastly different optimized prompt. However, both prompts result in very similar performance, despite solving the task in different ways. While the default initialization produces a prompt that leverages movie ratings and genre information from databases, the misleading prompt emphasizes concise explanations. Despite these contrasting approaches, both deliver comparable outcomes.

## G   PROMPT OPTIMIZATION PERFORMANCE IN VERY SMALL LANGUAGE MODEL : LLAMA-3.2-1B-INSTRUCT

While we primarily focused on two popular small language models - Llama-3-8B-Instruct and Gemma-2-9B-it, it is also interesting to see how GRᴇᴀTᴇʀ perform for even smaller LLM - namely Llama-3.2-1B-Instruct. Given that is one of the smallest modern language models, getting good task performance out of it just by optimizing prompt can be very useful for real world task deployment. As we can see in Table 8, GRᴇᴀTᴇʀ optimized prompts perform significantly better than prompts optimized by GPT-4 with other methods. This clearly shows the efficacy of GRᴇᴀTᴇʀ in prompt optimization even with very small language models.

Table 9: Optimized Prompts for LLama-3-8B-Instruct on GSM8K dataset.

| Method | Optimized Prompt | Score |
|---|---|---|
| TextGrad | You will answer a mathematical reasoning question. Think step by step. The last line of your response should be of the following format: 'Answer: VALUE' where VALUE is a numerical value. | 78.5 |
| APE | Work in sequence: Complete each task in order, tackling one task at a time, and only moving on to the next once it's finished. | 79.9 |
| APO | Break down complex problems into smaller, logical steps, considering mathematical operations, variable relationships, and implicit rules. Provide a clear, sequential solution, accounting for nuanced language and context. | 81.1 |
| PE2 | Break down complex problems into smaller, manageable steps, and solve them step by step. | 80.1 |
| GREATER | Use your knowledge reasoning and think step by step. Finally give the actual correct answer. | 82.6 |

## H  PROMPT OPTIMIZATION RESULTS: LLAMA-3-8B-INSTRUCT

### H.1  GSM8K AND FOLIO: OPTIMIZED PROMPTS

In Table 9 and Table 10, we first show the optimized prompts with Llama-3-8B-Instruct on GSM8K and FOLIO dataset respectively. As we can see, GREATER performs on par or better than all the baselines in every scenario.

Table 10: Optimized Prompts for LLama-3-8B-Instruct on FOLIO dataset.

| Method | Optimized Prompt | Score |
|---|---|---|
| TextGrad | You will answer a reasoning question by identifying the essential information, making specific conclusions, and providing nuanced and detailed reasoning. Think critically and systematically, focusing on the most relevant details, and avoid unnecessary complexity... | 56.2 |
| APE | Tackle it incrementally! | 57.6 |
| APO | Analyze the premises step by step, identifying specific details, assumptions, and ambiguities. Draw a logical conclusion based on the evidence provided, considering multiple perspectives and potential counterarguments, while accounting for scope, context, and edge cases. | 58.6 |
| PE2 | Analyze the statement based on the provided premise, determining whether it is true, false, or uncertain. Consider all relevant information to reach a logical conclusion. | 62.6 |
| GREATER | Use of logical deductions to show if your conclusion matches an appropriate option you chose from multiple options above by explaining how to determine whether the given conclusion follows from the given information above by explaining each step taken during the process. | 62.6 |

### H.2  BIG BENCH HARD (BBH): OPTIMIED PROMPTS AND DETAILED RESULTS

Figure 5 shows that GreaTer outperforms state-of-the-art (SOTA) prompt optimization methods, including APO (Pryzant et al., 2023), TextGrad (Yuksekgonul et al., 2024), APE (Zhou et al., 2022), and PE2 (Ye et al., 2023), across 71.4% to 85.% of tasks. A full breakdown of these results is provided in Figure 6. Upon closer inspection, it becomes evident that GREATER consistently matches or surpasses the performance of other SOTA methods, highlighting the robustness and reliability of our approach. In contrast, other methods show markedly less consistency due to their sole reliance on LLM judgment.

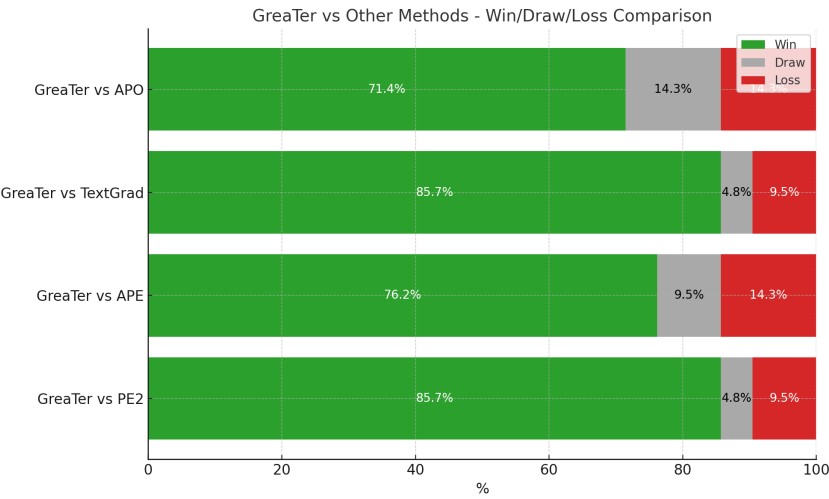

Figure 5: Win/Draw/Loss Comparison of GREATER and SOTA prompt optimization techniques APO, TextGrad, APE, and PE2 in optimization with Llama-3-8B-Instruct. GREATER maintains a significant winning margin over these methods, highlighting its effectiveness in optimization.

Finally, Table 11 presents all the optimized prompts generated by the baseline methods and our proposed approach. TextGrad prompts are truncated due to their excessive length, making them challenging to fit on a page, which may also explain their inconsistent performance.

Table 11: List of optimized prompts for BBH tasks by different prompt optimization methods with Llama-3-8B-Instruct.

| Method | Optimized Prompt |
|---|---|
| | formal_fallacies |
| PE2 | Determine the validity of the given argument. |
| APE | Simplify and analyze. |
| TextGrad | You will answer a reasoning question by explicitly identifying the key relationships between the premises and the conclusion, and explaining how they lead to the conclusion. Use clear and concise language to facilitate understanding, and... |
| APO | Analyze the argument step by step, considering premises, logical connections, and conditional statements. Identify the conclusion and evaluate its validity, considering sufficient and necessary conditions, counterexamples, and alternative scenarios. |
| GREATER | Use formal notation and and think step by step. Finally give the actual correct answer. |
| | salient_translation_error_detection |
| PE2 | Identify the type of error in the translation from German to English. n nSource: [insert source text] nTranslation: [insert translation] nError type: [one of the following] n(A) Modifiers or Adjectives n(B) Numerical Values n(C) Negation... |
| APE | Clarify your thoughts, break it down step by step. |
| TextGrad | You will answer a reasoning question by providing a detailed analysis of the original text and the translation. Think step by step, considering multiple possible explanations for the error. Clearly explain how each step leads... |

**Continued on next page**

| Method | Optimized Prompt |
|---|---|
| APO | Analyze the translation error by carefully reading the original sentence and identifying the specific mistake. Consider the exact words, phrases, and grammatical structures to determine the correct error type from the options. |
| GREATER | Use your answer reasoning as if I had step. I would be taking correct answer. |

| | tracking_shuffled_objects_five_objects |
|---|---|
| PE2 | Let's think step by step. |
| APE | Take it one step at a time: Focus on one task, complete it, then move on to the next. |
| TextGrad | You will answer a reasoning question by providing a step-by-step breakdown of the process. Use vivid and descriptive language to describe the events, and make sure to highlight the key connections and relationships between each... |
| APO | Let's think step by step. |
| GREATER | Use this process as an explanation stepwise for each step until you get to as given above Alice has got originaly the following as follows. |

| | causal_judgement |
|---|---|
| PE2 | What action(s) led to the outcome? Let's break it down step by step. |
| APE | Break down your thinking into clear, consecutive steps. |
| TextGrad | You will answer a reasoning question by explicitly connecting the events and outcomes, considering multiple perspectives and potential counterarguments, and providing nuanced explanations that take into account the context in which the events occurred. Think... |
| APO | Analyze the situation by identifying the direct and indirect causes, considering multiple perspectives, and evaluating counterfactuals. Provide a clear and concise answer, taking into account the context and nuances of the situation. Focus on the... |
| GREATER | Use causal diagram. The correct option ask about whether there the variable C of about whether a specific cause is sufficient. The answer a causal relationship between C to D if the probability P that C occurs given E changes. |

| | boolean_expressions |
|---|---|
| PE2 | Evaluate logical expressions step by step, considering the order of operations and specific values. Break down expressions into parts, and evaluate each part using 'or', 'and', and 'not' rules. |
| APE | Analyze and simplify. |
| TextGrad | You will answer a reasoning question by breaking down the expression into smaller, manageable parts. Provide a concise and clear explanation, using precise and concise language to describe the logical operations used to arrive at... |
| APO | Evaluate the boolean expression by following PEMDAS and applying boolean logic rules (AND, OR, NOT). Handle parentheses carefully. Consider edge cases and provide a step-by-step explanation of your reasoning, including any assumptions made. |
| GREATER | Use this statement with a conditional if know what is the value True of and what Not False means. Or not True and also boolean. In explain your. |

| | object_counting |
|---|---|
| PE2 | Let's think step by step. |
| APE | Break it down, step by step. |

**Continued on next page**

| Method | Optimized Prompt |
|---|---|
| TextGrad | You will answer a reasoning question about counting objects. Think step by step, considering the context of the question and using it to inform your answer. Be explicit in your counting process, breaking it down... |
| APO | Let's think step by step. |
| GREATER | Use only addition. Add think step by step. Finally give the actual correct answer. |

| | navigate |
|---|---|
| PE2 | Check if the instructions return to the starting point by calculating the total number of steps taken. |
| APE | Clarify your thoughts, analyze step by step. |
| TextGrad | You will answer a reasoning question by breaking down the problem step-by-step and providing explicit explanations for each step. Think carefully about the instructions and consider alternative scenarios. Use clear and precise language to describe... |
| APO | Analyze the instructions step by step, considering each action's effect on your position. Use logical reasoning to determine if you return to the starting point. |
| GREATER | Use your reasoning here. I would like numbers assigned.. to.. To represent moving. |

| | sports_understanding |
|---|---|
| PE2 | Assess the plausibility of the sentence. Is it likely to be true or fictional? |
| APE | Break down the task into manageable parts, examining each element thoroughly. |
| TextGrad | You will answer a reasoning question by providing a clear and concise step-by-step breakdown of your thought process, focusing on the most relevant and concrete evidence to support your claims. Consider alternative explanations and counterarguments... |
| APO | Assess the plausibility of the sentence, considering both literal and figurative meanings, as well as context and domain knowledge. Evaluate the sentence's coherence and relevance to the given context. |
| GREATER | Use the context or a sentence similar prior knowledge. Assume you a journalist, I would have been covering NHL hockey in Minnesota before joining this assignment to report sports. |

| | reasoning_about_colored_objects |
|---|---|
| PE2 | Analyze the input and options step by step to identify the correct answer. |
| APE | Break down into simpler components. |
| TextGrad | You will answer a reasoning question by carefully analyzing the problem statement, identifying the relevant information, and using logical deductions to arrive at a solution. Use precise and accurate language to describe your thought process,... |
| APO | Let's think step by step. |
| GREATER | Use this problem type as inspiration! which option best represents amu, the answer of all my are known. |

| | multistep_arithmetic_two |
|---|---|
| PE2 | Evaluate step-by-step and provide the correct answer, following the order of operations (PEMDAS). |
| APE | Decompose and analyze each part carefully. |

| Method | Optimized Prompt |
|---|---|
| TextGrad | You will answer a reasoning question by providing a clear, step-by-step breakdown of your thought process, using simple language and avoiding ambiguity. Focus on the key steps and simplify the intermediate calculations. Use descriptive variable... |
| APO | Evaluate the expression by following PEMDAS, handling parentheses, and accurately calculating with negative numbers. Break down complex expressions into simpler steps and provide the final answer. |
| GReaTer | Use PEMAS reasoning here and step by the. STEP to the actual number result and explain what PEMAS means by each step of how I would evaluate this expression correctly according follow these step wise... |

| date_understanding | |
|---|---|
| PE2 | Let's think step by step. |
| APE | Analyze step-by-step. |
| TextGrad | You will answer a reasoning question by breaking it down into manageable steps, focusing on simplicity and clarity in your reasoning. Provide a concise and clear explanation of your thought process, avoiding unnecessary conversions and... |
| APO | Let's think step by step. |
| GReaTer | Use the date today which will not would give us an error. solution is given as answer date is correct the option data and the current month and year to get to previous and current month of year to determine what the current data will look. |

| ruin_names | |
|---|---|
| PE2 | Identify the humorous edit of this artist or movie name. Choose an option that cleverly replaces a word or plays on words with the original name. Options may include the correct answer. |
| APE | Break it down, step by step. |
| TextGrad | You will answer a reasoning question by providing a step-by-step analysis of the options, highlighting the unique features and characteristics of each humorous edit. Consider the linguistic and cognitive factors that contribute to humor, such... |
| APO | Imagine a creative reinterpretation of the original name. Think outside the box and come up with a clever edit that's unexpected yet amusing. Consider tone, context, and audience when selecting the most humorous and engaging... |
| GReaTer | Use your logical reasoning to make this, not brute force checking.CONTEXT is provided below. |

| movie_recommendation | |
|---|---|
| PE2 | Find a movie that shares similar elements with the given films, considering narrative structure, memorable characters, genre blending, strong protagonists, and emotional impact. |
| APE | Calmly analyze, think critically. |
| TextGrad | You will answer a reasoning question by analyzing the given movies and identifying the most suitable match. Think step-by-step, focusing on the most distinctive features that connect the input movies, such as unique plot twists,... |
| APO | Analyze the movies' tone, genre, and style, considering action, drama, and comedy elements. Identify the most fitting movie from the options that shares these characteristics, focusing on overall themes and elements rather than individual features. |

**Continued on next page**

| Method | Optimized Prompt |
|--------|------------------|
| GREATER | Use movie ratings data available here above movies for reference. ThisHOFF has an interesting analysis based solely options to options based movies ratings expect from the other movies you are asked ones mentioned here you... |

| | **web_of_lies** |
|--------|------------------|
| PE2 | Evaluate statements about the truthfulness of others in a chain of lies or truth-telling. Determine if speakers are telling the truth or lying, considering each statement and the speaker's integrity. |
| APE | Let's take it one step at a time: analyze the task into smaller, manageable chunks, and then tackle each chunk individually to achieve a clear and focused approach. |
| TextGrad | You will answer a reasoning question by specifying the scope of 'the truth' and using explicit language to connect each step in your reasoning. Focus on essential steps and consider alternative perspectives. Use direct and... |
| APO | Analyze each statement individually, considering the speaker's truthfulness and potential contradictions. Determine the truth or falsehood of each statement, then use this information to evaluate the final statement. |
| GREATER | Use only statement reasoning.Let step ick. We need to step out from here to figure this one out step out step out step out step out from each of those. |

| | **disambiguation_qa** |
|--------|------------------|
| PE2 | Identify the antecedent of the pronoun, considering sentence structure and context. If ambiguous, provide evidence to support your answer. |
| APE | Clarify thoughts, analyze step by step. |
| TextGrad | You will answer a reasoning question by providing a step-by-step explanation of your thought process, considering the context, syntax, and semantics of the sentence, as well as the relationships between the entities mentioned. Use linguistic... |
| APO | Analyze the sentence and identify the antecedent of the pronoun. Consider the context, relationships between entities, and potential ambiguity. Provide a clear explanation for your answer, highlighting any relevant details that support your conclusion. |
| GREATER | Use is possible reasoning for either answer by step. Finally, the actual correct answer may also not have an explicit mention of |

| | **logical_deduction_five_objects** |
|--------|------------------|
| PE2 | Determine the correct order of objects based on logical relationships and statements provided. |
| APE | Break down and examine each stage carefully. |
| TextGrad | You will answer a reasoning question by breaking down the information into clear and concise steps. Use specific and unambiguous language to describe the relationships between the objects. Consider using diagrams or illustrations to help... |
| APO | Carefully analyze each statement, considering relationships between objects and logical implications. Eliminate options that contradict the statements. Recognize and resolve contradictions. Consider word order and syntax to ensure accurate conclusions. |
| GREATER | Use elimination logical reasoning and think step by step. Finally give the actual correct answer. |

| | **snarks** |
|--------|------------------|

| Method | Optimized Prompt |
|---|---|
| PE2 | Identify the sarcastic statement and explain the irony, mocking tone, and intended meaning. Consider language that is ironic, mocking, or opposite of what is meant. |
| APE | Break down the task into smaller steps, and let's tackle each one individually. |
| TextGrad | You will answer a reasoning question by considering multiple factors and providing a detailed, step-by-step analysis. Think critically about the context, speaker's intent, and audience's perspective. Pay particular attention to the tone and language used... |
| APO | Let's think step by step. |
| GReaTer | Use your common reasoning and judgment, by step. Finally give the actual correct answer. |
| | geometric_shapes |
| PE2 | Identify the quadrilateral or geometric shape drawn by this SVG path element. |
| APE | Analyze and simplify. |
| TextGrad | You will answer a reasoning question by analyzing the path's overall shape, examining how the individual segments contribute to the path's geometry, and provide more context and domain-specific knowledge about SVG path elements, such as... |
| APO | Analyze the SVG path element, focusing on both line segments and curves. Identify the starting and ending points, and recognize patterns in the movement. Consider the overall path structure and geometric properties to determine the... |
| GReaTer | Use your best answer from the I. answer the options. assistantactiveassistance-sassistantative be a mathematical object with vertices. If there be represented by the path. |
| | hyperbaton |
| PE2 | Identify the correct adjective order in the given sentence. Adjectives typically follow a specific order: opinion, shape, size, material, etc., with exceptions and context-dependent variations. |
| APE | Organize your ideas, simplify them. |
| TextGrad | You will answer a reasoning question. Think step by step. Provide explicit explanations for each step. Consider breaking down complex concepts into smaller, more manageable parts. When analyzing the sentence, pay close attention to the... |
| APO | Analyze the adjective order in each sentence, considering context, typical order of opinion, adverb role, and exceptions. Provide the correct sentence with adjectives in the most natural and idiomatic order. |
| GReaTer | Use the reasoning and examples you would step. Finally give the actual correct answer. |
| | penguins_in_a_table |
| PE2 | Count step by step and find the answer. |
| APE | Unpack your ideas, review thoroughly. |
| TextGrad | You will answer a reasoning question by following a structured approach. Think step by step, considering the most critical information and alternative explanations. Use precise language and clarify the scope of the question. Organize your... |
| APO | Let's think step by step. |

**Continued on next page**

| Method | Optimized Prompt |
|---|---|
| GREATER | Use this to solve this puzzle step by step. Finally give the actual correct answer. |
| | temporal_sequences |
| PE2 | Find the time windows when the person was not busy or occupied to visit the location, considering their schedule. |
| APE | Dissect and analyze the information. |
| TextGrad | You will answer a reasoning question by identifying the most plausible answer, explicitly stating assumptions and considering alternative explanations. Clearly explain how each piece of evidence supports your conclusion, and provide specific and precise language... |
| APO | Let's think step by step. |
| GREATER | Use the timeline provided and answer step by step. Finally give the actual correct answer. |

# I  PROMPT OPTIMIZATION RESULTS: GEMMA-2-9B-IT

## I.1  GSM8K AND FOLIO: OPTIMIZED PROMPTS

In Table 12 and Table 13, we first show the optimized prompts with Llama-3-8B-Instruct on GSM8K and FOLIO dataset respectively. As we can see, GREATER performs on par or better than all the baselines in every scenario.

Table 12: Optimized Prompts for Gemma-2-9B-it on GSM8K dataset

| Method | Optimized Prompt | Score |
|---|---|---|
| TextGrad | You will answer a mathematical reasoning question. Think step by step. | 87.8 |
| APE | Let's think step by step. | 88.6 |
| APO | Let's think step by step. | 88.6 |
| PE2 | Let's think step by step. | 88.6 |
| GREATER | Use these logical reasoning process steps and explain Step. step. Here is correct answer. | 89.4 |

Table 13: Optimized Prompts for Gemma-2-9B-it on FOLIO dataset

| Method | Optimized Prompt | Score |
|---|---|---|
| TextGrad | You will answer a reasoning question. Think step by step, carefully considering all provided information and identifying any potential contradictions or ambiguities. When evaluating statements about preferences, . . . | 67.5 |
| APE | Divide the problem into manageable chunks. | 67.5 |
| APO | (empty prompt) | 63.1 |
| PE2 | Given the premises, determine the certainty of the following statement. Choose from: * Conclusive True * Conclusive False * Uncertain | 62.1 |
| GREATER | Use logic or reasoning and think step by step. Finally give the actual correct answer. | 68.5 |

## I.2  BIG BENCH HARD (BBH): OPTIMIZED PROMPTS AND DETAILED RESULTS

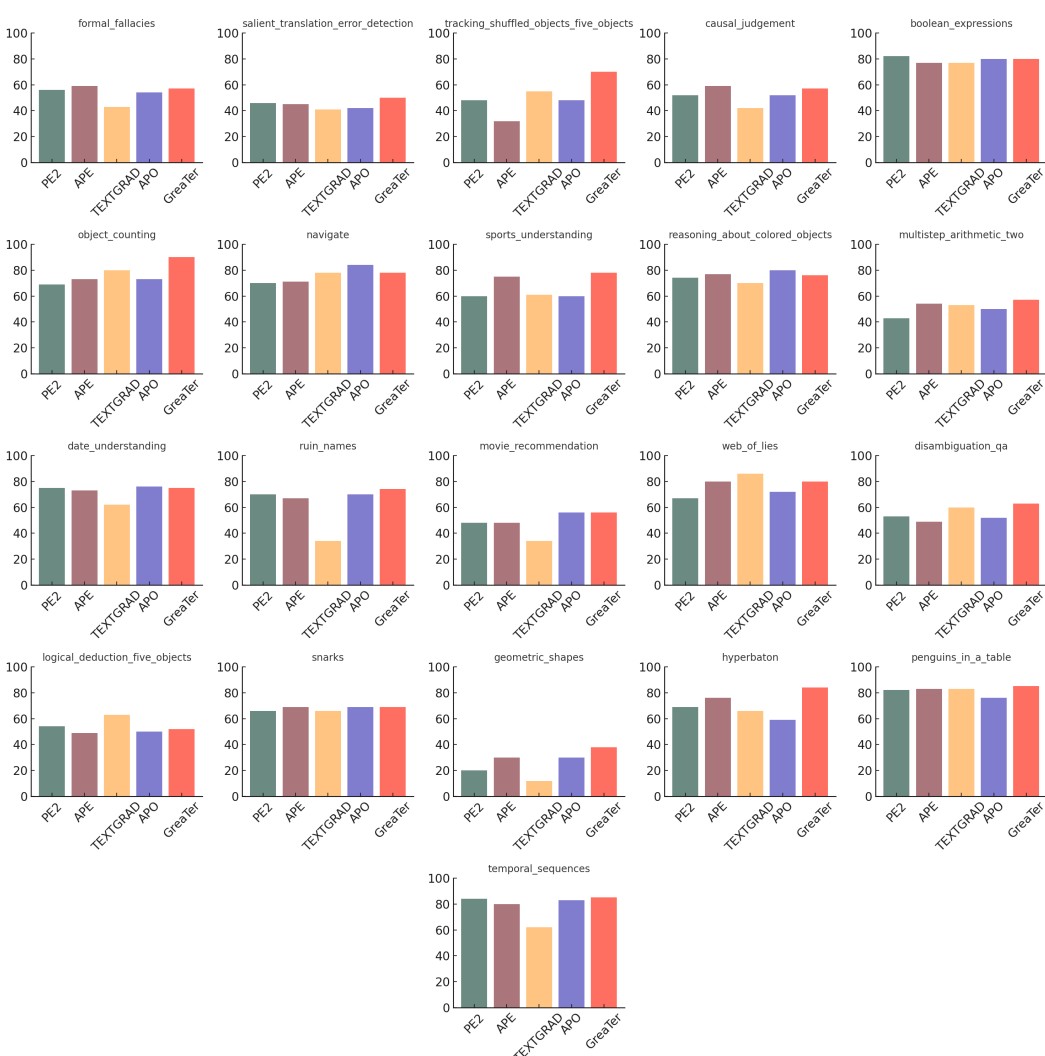

Figure 6: Full performance breakdown across 21 BBH tasks of GREATER and SOTA prompt optimization techniques APO, TextGrad, APE, and PE2 in optimization with Llama-3-8B.

Table 14: List of optimized prompts for BBH tasks by different prompt optimization methods with Gemma-2-9B-it.

| Method | Optimized Prompt |
|---|---|
| | multistep_arithmetic_two |
| PE2 | Let's think step by step and calculate the result. |
| APE | Let's think step by step. |
| TextGrad | You will answer a reasoning question. Remember to follow the order of operations (PEMDAS/BODMAS) when solving the problem step-by-step. Think step-by-step, clearly outlining each operation you perform. Begin by simplifying any expressions within parentheses. Then,... |
| APO | Let's think step by step. |
| GREATER | Use parentheses, and and the step wise order. Solve for the correct answer. |

**Continued on next page**

| Method | Optimized Prompt |
|---|---|
| **reasoning_about_colored_objects** | |
| PE2 | Let's think step by step to determine the answer. |
| APE | Break this down into smaller, easier-to-handle sections. |
| TextGrad | You will answer a reasoning question. Your goal is to determine the answer to the question based on the provided information and explain your thought process clearly. Present your reasoning in the most concise and... |
| APO | Analyze the given text and answer the question. Provide a brief explanation of your reasoning, listing the steps you took. |
| GREATER | Use your logic. Please answer. person. Yout answer. A B. |
| **geometric_shapes** | |
| PE2 | Analyze the SVG path data in the 'd' attribute and identify the most specific geometric shape it represents, considering commands like 'M', 'L', and others. |
| APE | Break this down into smaller parts. |
| TextGrad | You will analyze the provided SVG path element and determine the shape it represents. Consider the number of line segments (L commands) and their connections to identify the shape. Look for patterns in the coordinates... |
| APO | (empty) |
| GREATER | Use an logical reasoning and think step by step. Finally give the actual correct answer. |
| **sports_understanding** | |
| PE2 | Let's think step-by-step and assess the plausibility of the following sentence. |
| APE | Can we break this down into smaller steps? |
| TextGrad | You will evaluate the plausibility of statements based on the provided context and established rules and mechanisms of football. When evaluating plausibility, consider the relationship between the statement and the broader context of a football... |
| APO | Evaluate the plausibility of this sentence, considering both general knowledge and the context of sports. Think about whether such an event is realistically possible. |
| GREATER | Use your understanding to explain the step by step. Finally give the actual correct answer. |
| **disambiguation_qa** | |
| PE2 | Let's think step by step. Select the option that correctly identifies the antecedent of the pronoun. |
| APE | Walk me through this process step by step. |
| TextGrad | You will answer a reasoning question. Think step by step, paying close attention to the grammatical structure of the sentence and identify the function of each word. When encountering pronouns, clearly explain which noun or... |
| APO | Identify the noun or phrase that the pronoun 'they' refers to in each sentence. Choose the most specific and accurate antecedent. If ambiguous, select 'Ambiguous'. |
| GREATER | Use of logical connection instead think step by step. Finally give the actual correct answer. |
| **ruin_names** | |

| Method | Optimized Prompt |
|--------|------------------|
| PE2 | Let's think step by step. |
| APE | Let's think step by step. |
| TextGrad | You will analyze humorous edits of artist or movie names, assuming your audience is [specify target demographic]. \n\n**Key Humor Components:**\n\n* **Incongruity:** Juxtaposing clashing or unexpected elements.\n* **Surprise:** Unexpected twists or... |
| APO | Identify the **most humorous** edit of the given artist or movie name. Focus on **creative wordplay** and **unexpected twists**, not just phonetic changes. |
| GREATER | Use your logic, not change spelling. punny play with existing names to get the answer. |

| | hyperbaton |
|--------|------------------|
| PE2 | Let's identify the sentence with the incorrect adjective order: |
| APE | Explain each step separately. |
| TextGrad | Your primary goal is to clearly and accurately explain the reasoning behind the correct answer. First, discuss the relevant grammatical principles at play when arranging adjectives in a sentence. Then, apply these principles to the... |
| APO | Let's think step by step. |
| GREATER | Use your knowledge to and think step by step. Finally give the actual correct answer. |

| | causal_judgement |
|--------|------------------|
| PE2 | Analyze the scenario and determine if the person's action was a direct cause of the event. Explain your reasoning. |
| APE | Divide the problem into manageable chunks. |
| TextGrad | You will answer a causation question, demonstrating a nuanced understanding of cause-and-effect relationships. Consider complex interactions between multiple factors, analyze situations with indirect or delayed effects, and evaluate the role of probability and likelihood in... |
| APO | What single action was the most immediate cause of the stated outcome? |
| GREATER | Use proper causal reasoning . step through step. Finally give the actual correct answer. |

| | boolean_expressions |
|--------|------------------|
| PE2 | Evaluate the truth value of the following logical expression, showing your reasoning step-by-step: |
| APE | Let's think step by step. |
| TextGrad | You will answer a reasoning question involving logical expressions. Analyze the problem logically, providing a thorough explanation of how each logical operator (and, or, not) influences the truth value of the expression. Use clear examples... |
| APO | Let's think step by step. |
| GREATER | Use logical truth tables. method or truth operations. And. Or or The. following statements evaluate each. |

| | object_counting |
|--------|------------------|
| PE2 | Let's think step by step. |
| APE | Let's think step by step. |

**Continued on next page**

| Method | Optimized Prompt |
|---|---|
| TextGrad | You will answer a reasoning question. Clearly calculate the answer to the question. \n\nList all the relevant elements involved in the calculation. Then, using those elements, perform the calculation to find the answer.... |
| APO | Let's think step by step. |
| GREATER | Use counting maths . I see that is more that a single. I have more then ten because there more the number. |

| | movie_recommendation |
|---|---|
| PE2 | Let's identify a movie with a similar genre to Braveheart, Dances with Wolves, Pulp Fiction, and Schindler's List. |
| APE | Break this down into smaller, easier-to-handle parts. |
| TextGrad | You will answer a reasoning question by identifying the movie most similar to a given set. To arrive at your answer, follow these steps:\n\n1. **Analyze each movie:** Identify and analyze specific plot points,... |
| APO | Classify the movie option most similar in genre to the given film list. Choose the best fit. |
| GREATER | Use only reasoning and reasoning based logic. I chose option. I think the film that fits the listed criteria but is more readily avaliabke on common viewing services. |

| | formal_fallacies |
|---|---|
| PE2 | Let's think step by step. |
| APE | Let's think step by step. |
| TextGrad | You will answer a reasoning question. Your task is to determine if the conclusion *logically follows* from the premises, regardless of whether the conclusion is true in the real world. Think step-by-step and clearly articulate... |
| APO | Let's think step by step. |
| GREATER | Use modus ponenis incorrectly because step is incorrect for some premises. Invalid due because. |

| | salient_translation_error_detection |
|---|---|
| PE2 | Let's think step by step.\nIdentify the error type in these translations: Named Entities, Numerical Values, Modifiers or Adjectives, Negation or Antonyms, Facts, or Dropped Content. |
| APE | Break this down into smaller steps. |
| TextGrad | You will answer a reasoning question based on a text passage. Carefully compare the source text and the provided translation, paying close attention to the meaning of individual words and phrases. Identify any words or... |
| APO | Remember, a good prompt for a zero-shot classifier should be:\n\n* **Clear and concise:** Avoid ambiguity and unnecessary jargon.\n* **Specific:** Clearly define the task and the expected output format.\n* **Grounded in the... |
| GREATER | Use the following based on this information, using a specific error category as an example. |

| | penguins_in_a_table |
|---|---|
| PE2 | Let's think step by step to answer the following question: |
| APE | Let's think step by step. |

| Method | Optimized Prompt |
|---|---|
| TextGrad | Your task is to answer a reasoning question by carefully analyzing the provided information. Pay close attention to the specific details and facts presented in the text. Identify the key pieces of information that are... |
| APO | Let's think step by step. |
| GReaTer | Use the the provided context, ,,and explaining. The answer and explain the solution is process in a simple step. step guide for someone just leering about coding Python. |

| | tracking_shuffled_objects_five_objects |
|---|---|
| PE2 | Let's trace the changes in partners step-by-step to determine the final state. |
| APE | Explain it step by step. |
| TextGrad | Your goal is to determine the final state of a given scenario by carefully analyzing a series of steps. Pay close attention to each step and track how items or values change hands. After detailing... |
| APO | Let's think step by step. |
| GReaTer | Use logic series or process or best method this. Following each series. |

| | date_understanding |
|---|---|
| PE2 | Let's think step by step. |
| APE | Let's think step by step. |
| TextGrad | You will answer a reasoning question. Think step by step, paying close attention to any date formats presented in the question. Ensure your reasoning clearly reflects how you interpret and manipulate dates based on their... |
| APO | Let's think step by step. |
| GReaTer | Use your format Excel formula for this answer to find it . It have gotten some. |

| | web_of_lies |
|---|---|
| PE2 | Let's think step by step to determine the answer. |
| APE | Explain each step individually. |
| TextGrad | You will answer a reasoning question. Analyze the information carefully and identify the key relationships and deductions that lead to the solution. Express your reasoning concisely, highlighting the most important connections. Use clear and direct... |
| APO | Let's think step by step. |
| GReaTer | Use proper logical reasoning and think step by step. Finally give the actual correct answer. |

| | snarks |
|---|---|
| PE2 | Let's think step by step. Identify the most sarcastic statement. |
| APE | Decompose the problem into manageable subtasks. |
| TextGrad | You will answer a reasoning question. Think step by step. The last line of your response should be of the following format: 'Answer: $VALUE' where VALUE is a numerical value. |
| APO | Let's think step by step. |
| GReaTer | Use a logical reasoning and think step by step. Finally give the actual correct answer. |

| Method | Optimized Prompt |
| --- | --- |
| | temporal_sequences |
| PE2 | Given the following information about [person's name]'s day, determine the time slot(s) when they could have gone to the coffee shop, which closes at 7pm. |
| APE | Walk me through the process, step by step. |
| TextGrad | You will answer a reasoning question. Break down the problem into smaller steps, identifying key pieces of information and eliminating possibilities based on the given facts. Present your reasoning in a clear, step-by-step manner, explicitly... |
| APO | (empty) |
| GREATER | Use process logic, and eliminate options by considering what we do the actual correct answer. |
| | logical_deduction_five_objects |
| PE2 | Let's think step-by-step to determine the position of the specified object within the sequence. |
| APE | Let's think step by step. |
| TextGrad | Your goal is to determine the position of a specific item within a described arrangement. You will be presented with a set of statements describing the arrangement and a question about the position of a... |
| APO | Let's think step by step. |
| GREATER | Use elimination process, use this information, to eliminate choices sufficient information, eliminate to the. correct answer choice correct. |
| | navigate |
| PE2 | Let's think step-by-step and determine your final position relative to the starting point based on these instructions. |
| APE | Let's think step by step. |
| TextGrad | You will answer a reasoning question involving changes in position or state. \n* For each movement, clearly state the direction (e.g., '3 steps to the left') along with the number of steps.\n* Assume... |
| APO | Let's think step by step. |
| GREATER | Use proper mathematical logic, explaining step by step. Finally give the actual correct answer. |

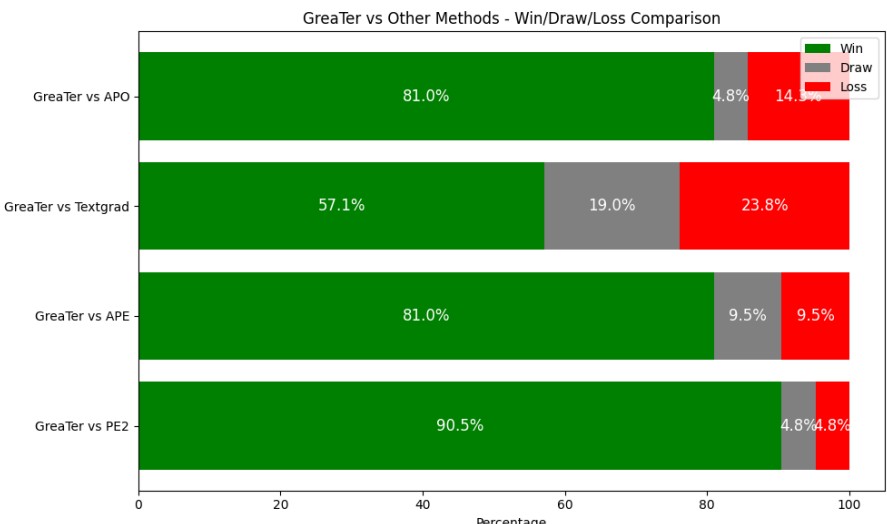

Figure 7: Win/Draw/Loss Comparison of GREATER and SOTA prompt optimization techniques APO, TextGrad, APE, and PE2 in optimization with Gemma-2-9B-it. GREATER maintains winning margin over these methods, highlighting its effectiveness in optimization.

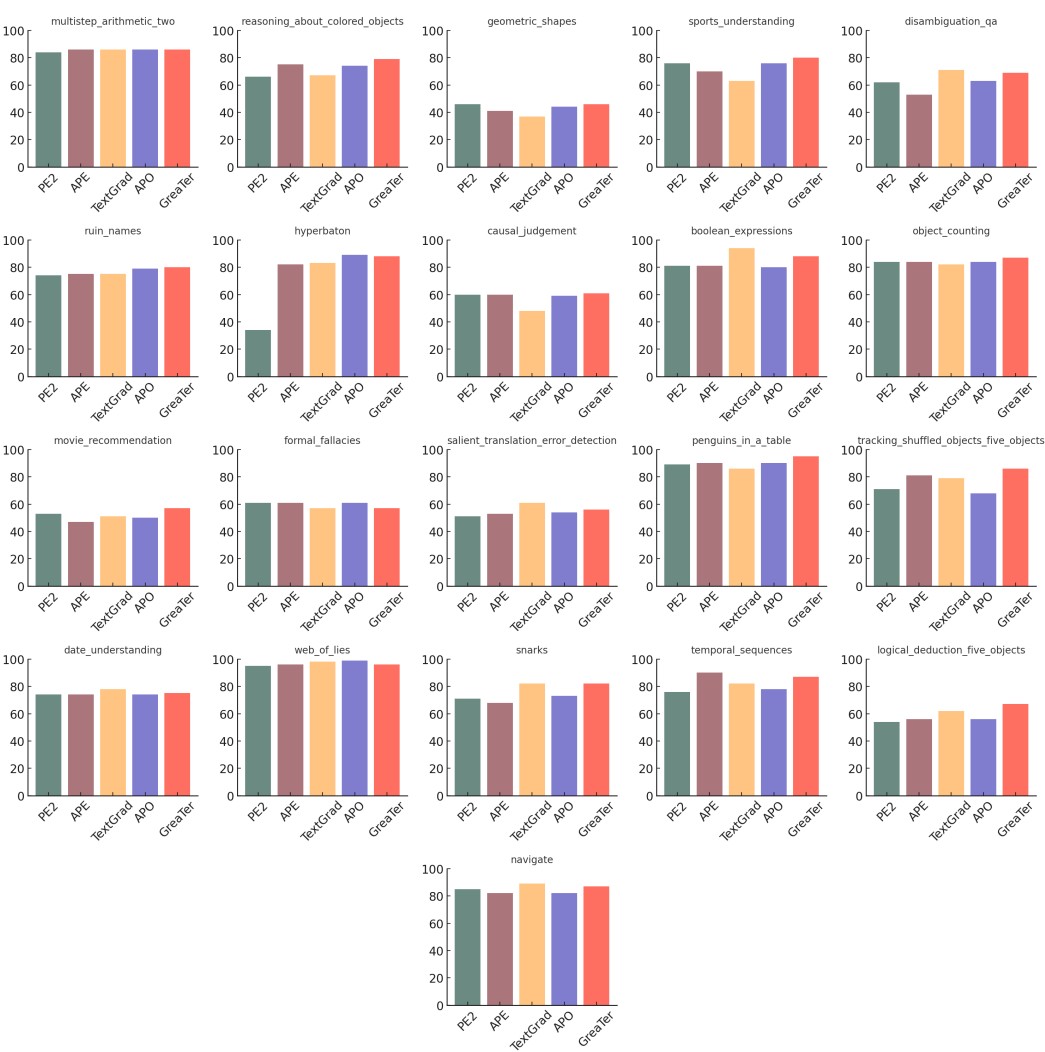

Figure 8: Full performance breakdown across 21 BBH tasks of GREATER and SOTA prompt optimization techniques APO, TextGrad, APE, and PE2 in optimization with Gemma-2-9B.

