# OpenReview forum: "GReaTer: Gradients Over Reasoning Makes Smaller Language Models Strong Prompt Optimizers"
_ICLR.cc/2025/Conference — ICLR 2025 Poster_

### Official Review · Reviewer_7nfg · 2024-11-03

**Soundness:** 3
**Presentation:** 2
**Contribution:** 3
**Rating:** 6
**Confidence:** 4

**Summary:**

This paper proposed a novel prompt optimization method that leverages some small language models to generate reasoning/explanation for problem solutions and utilizes the gradients on probable token candidates from the loss of answer tokens over reasoning tokens to refine the prompt optimization process. Experiments over 2 models and several benchmarks demonstrate that this method is effective.

**Strengths:**

- The idea of applying gradients over reasoning tokens to refine the selection of prompt tokens is novel.
- The proposed method has shown empirical success.

**Weaknesses:**

- Presentation is unclear:
    - Slight notation abuse: the output space of $f_{\text{LLM}}$ is not well-defined. Does it output texts or token probabilities? They are different in Eq. 2&3.
    - It is not clear for the main results presented in Sec. 5.2 which model is used for prompt evaluation and which model is used for optimizing the prompt (e.g., in Table 2).
    - The complete algorithm 1 in Appx. B.1 should be moved to the main text for better understanding, as the presentation of the designed method is not very easy to follow.

- Sec. 5.6 does not show much insight which I think should be omitted from the main text.
- How to justify “simply considering fLLM(y|x ⊙ p) would give us the wrong objective to optimize, which in turn will give incorrect gradient information” in lin 237-238? The idea of taking gradients over reasoning paths is not well-explained. Intuitively, I expect the distribution of fLLM(y|x ⊙ p) to be more skewed than fLLM(y|x ⊙ p ⊙ r) as we can view $r$ as more computation to search for the right answer $y$.
- What does the gradient direction $\frac{\partial \mathcal{L}}{\partial \epsilon_i}$ physically mean? Why is there a negative sign and why is the token with the highest negative gradient value selected instead of the one with the largest gradient norm?
- Recent literature [1, 2, 3] related to applying the gradient concept in prompt optimization should be at least discussed.

[1] Pryzant, R., Iter, D., Li, J., Lee, Y. T., Zhu, C., & Zeng, M. (2023). Automatic prompt optimization with" gradient descent" and beam search.

[2] Hu, W., Shu, Y., Yu, Z., Wu, Z., Lin, X., Dai, Z., ... & Low, B. K. H. (2024). Localized zeroth-order prompt optimization.

[3] Wen, Y., Jain, N., Kirchenbauer, J., Goldblum, M., Geiping, J., & Goldstein, T. (2024). Hard prompts made easy: Gradient-based discrete optimization for prompt tuning and discovery

**Questions:**

- what if there is no intersection in Eq.4? And how to guarantee semantic coherence among different token positions?
- Can you explain why the performance gain from gradient over reasoning is much smaller on Gemma model compared with LLaMA model?
- How about the performance on 1.5B or 3B model? 9B model is actually not very small in practice.

---

> ### Author Response · Authors · 2024-11-25
> **Author Response to Reviewer 7nfg (Part 1/3)**
>
> We are thankful for your detailed and comprehensive review! We highly appreciate your suggestions and have uploaded an updated PDF by including new additional experiments and improved paper presentation accordingly. Below we summarize and address your concerns:
>
> ---
>
> > ### **Weakness 1: Presentation is unclear**
> > - **Slight notation abuse**: The output space of $ f_{LLM} $ is not well-defined. Does it output texts or token probabilities? They are different in Eq. 2 & 3.
>
> We agree this notation was a little confusing. In **Equation 3**, we are using $ f_{LLM}(\cdot \mid \ldots) $ to take token probabilities. On the other hand, in Section 4.1, we have updated the manuscript to clarify that the reasoning chain $ r $ is sampled from $ f_{LLM}(x \odot p) $, i.e. $ r \sim f_{LLM}(x \odot p) $, and we have made changes accordingly in **Equation 2**. With the updates in Section 4.1, we hope this resolves the confusion.
>
> > - **It is not clear for the main results presented in Sec. 5.2 which model is used for prompt evaluation and which model is used for optimizing the prompt (e.g., in Table 2).**
>
> Thank you so much for pointing this out. To improve the presentation of **Table 2**, we have explicitly mentioned the target model (the model used for evaluating the prompt) and the optimizer model in the table heading. We also clarified this in the caption to improve readability. The table is summarized below:
>
> | **Target Model**      | **Method (Optimized by)**     | **GSM8K** | **movie_rec.** | **object_count.** | **tracking_five.** | **hyperbaton** | **causal** | **Average** |
> |------------------------|-------------------------------|-----------|----------------|-------------------|---------------------|----------------|-----------|------------|
> | **Llama-3-8B**         | APE (GPT-4)                  | 80.7      | 50             | 82                | 50                  | 76             | 56        | 62.8       |
> |                        | EvoPrompt (GPT-3.5)          | -         | 48             | 74                | 42                  | 68             | 48        | 56.0       |
> |                        | APO (GPT-4)                  | 81.1      | 56             | 68                | 49                  | 75             | 51        | 59.8       |
> |                        | PE2 (GPT-4)                  | 81.5      | 48             | 82                | 45                  | 79             | 49        | 60.6       |
> |                        | OPRO (PaLM-2-L)              | 82.3      | 60             | 78                | 40                  | 70             | **57**    | 61.0       |
> |                        | **GreaTer (Llama-3-8B)**       | **82.6**  | **57**         | **90**            | **70**              | **84**         | **57**    | **71.6**   |
> | **Gemma-2-9B**         | APE (GPT-4)                  | 89.2      | 48             | 61                | 83                  | 83             | 60        | 67.0       |
> |                        | EvoPrompt (GPT-3.5)          | -         | 51             | 70                | 82                  | 83             | 61        | 69.4       |
> |                        | APO (GPT-4)                  | 89.3      | 52             | 84                | 72                  | 82             | 59        | 69.8       |
> |                        | PE2 (GPT-4)                  | **89.6**  | 50             | 65                | 71                  | 84             | **64**    | 66.8       |
> |                        | OPRO (PaLM-2-L)              | 89.0      | 50             | 58                | 76                  | 81             | 58        | 64.6       |
> |                        | **GreaTer (Gemma-2-9B)**       | 89.4      | **56**         | **87**            | **85**              | **88**         | 61        | **75.4**   |
>
> > - **The complete Algorithm 1 in Appx. B.1 should be moved to the main text for better understanding, as the presentation of the designed method is not very easy to follow.**
>
> We really appreciate your feedback on improving the layout. As per your suggestions, we have moved **Algorithm 1** to the main text in Section 4, where the methodology of GreaTer is discussed.
>
> ---
>
> > ### **Weakness 2**
> > **Sec. 5.6 does not show much insight, which I think should be omitted from the main text.**
>
> We also agree with your suggestion to transfer **Section 5.6** (*Comparison against few-shot in-context learning*) to the Appendix. We believe this change greatly improves the overall readability of the paper.

---

> ### Author Response · Authors · 2024-11-25
> **Author Response to Reviewer 7nfg (Part 2/3)**
>
> > ### **Weakness 3**
> > **How to justify “simply considering $ f_{LLM}(y \mid x \odot p) $ would give us the wrong objective to optimize, which in turn will give incorrect gradient information” in lines 237-238? The idea of taking gradients over reasoning paths is not well-explained. Intuitively, I expect the distribution of $ f_{LLM}(y \mid x \odot p) $ to be more skewed than $ f_{LLM}(y \mid x \odot p \odot r) $, as we can view $ r $ as more computation to search for the right answer $ y $.**
>
> Thank you for seeking clarification regarding the significance of gradient over reasoning. As outlined in the manuscript, a core strength of GreaTer lies in its "Gradient Over Reasoning," as the reasoning chain effectively captures the task loss. To illustrate its importance more clearly, consider the following:
>
> - **Input, $ x $:** "The cafeteria had 23 apples. If they used 20 to make lunch and bought 6 more, how many apples do they have?"
> - **Prompt, $ p $:** "Use logical reasoning and think step by step."
> - **Reasoning chain, $ r $:** "The cafeteria had 23 apples originally. They used 20 to make lunch. So they had $ 23-20=3 $. They bought 6 more, so they now have $ 3+6=9 $."
>
> When we condition the language model to generate the answer $ y $, there are two distinct distributions to consider:
>
> 1. **$ f_{LLM}(y \mid x \odot p) $:**
>    This distribution is conditioned on the input $ x $ and the prompt $ p $, but without reasoning paths explicitly included. Here, the model sees:
>    *"The cafeteria had 23 apples. ... how many apples do they have? Use logical reasoning and think step by step. The final answer is ..."*
>
>    However, the logits at this stage do not incorporate the intermediate reasoning chain $ r $. This will lead to noisy or incorrect loss as well as gradient calculation for $ p $. The final output probabilities may misrepresent the likelihood of the correct answer due to the lack of explicit reasoning steps.
>
> 2. **$ f_{LLM}(y \mid x \odot p \odot r) $:**
>    This distribution is conditioned on $ x $, $ p $, and the reasoning chain $ r $ induced by the prompt $ p $. Here, the model sees:
>    *"The cafeteria had 23 apples. ... how many apples do they have? Use logical reasoning and think step by step ... The cafeteria had 23 apples originally. They used 20 to make lunch. … $ 3+6=9 $. Therefore, the final answer is ..."*
>
>    The presence of $ r $ ensures that $ y $ is conditioned on the generated reasoning chain induced by the prompt, representing accurate loss and therefore gradients with respect to $ p $. This represents not only the answer but also the induced reasoning.
>
> This also aligns with [1], as it shows that without generating reasoning steps, final answer probabilities might be incorrect.
>
> Lastly, we included additional experiments comparing GreaTer to other gradient-based approaches that do not consider the reasoning chain [2,3] in **Table 6** (Appendix C of the updated manuscript). The table is summarized below:
>
> | Method       | Llama-3-8B (movie_rec.) | Llama-3-8B (tracking_five) | Gemma-2-9B (movie_rec.) | Gemma-2-9B (tracking_five) |
> |--------------|--------------------------|----------------------------|--------------------------|----------------------------|
> | AutoPrompt   | 51                    | 38                       | 50                    | 73                       |
> | PEZ          | 42                    | 35                       | 45                    | 73                       |
> | **GreaTer**    | **56**                | **70**                   | **57**                | **86**                   |
>
> ---
>
> > ### **Weakness 4**
> > **What does the gradient direction $ \frac{\partial L}{\partial \epsilon_i} $ physically mean? Why is there a negative sign and why is the token with the highest negative gradient value selected instead of the one with the largest gradient norm?**
>
> Thank you so much for raising this important point. The gradient direction $ \frac{\partial L}{\partial \epsilon_i} $ indicates how much a token contributes to increasing the loss function. Since our goal is to reduce the loss, we focus on tokens with the highest negative gradient values, as they correspond to changes that decrease the loss most effectively. This is why we select the token with the most negative gradient value, rather than one with the largest gradient magnitude, which could signify either an increase or decrease in loss.

---

> ### Author Response · Authors · 2024-11-25
> **Author Response to Reviewer 7nfg (Part 3/3)**
>
> > ### **Weakness 5**
> > **Recent literature [3,4,5] related to applying the gradient concept in prompt optimization should be at least discussed.**
>
> Thank you so much for mentioning those works in the literature!
>
> As we discussed, Pryzant et al. [4] introduced the use of mini-batches of data to create natural language feedback, referred to as “natural language gradients.” These gradients critique the current prompt, mimicking the role of numerical gradients in optimization. This method, named APO, serves as one of our primary baselines across all experiments.
>
> Additionally, [5] highlighted that local optima are significantly more common than global optima in prompt optimization. Based on this observation, the authors proposed ZOPO, a method that incorporates a Gaussian process inspired by the Neural Tangent Kernel into zeroth-order optimization. This approach systematically explores local optima through iterative evaluations.
>
> Finally, Wen et al. [3] introduced the PEZ algorithm, a gradient-based discrete optimization approach that utilizes gradients without considering reasoning paths. As noted in Weakness 3, we compared our approach, GreaTer, against PEZ and observed a significant performance improvement from our method.
>
> We have incorporated these discussions into the related works section of the revised manuscript.
>
> ---
>
> > ### **Question 1**
> > **What if there is no intersection in Eq. 4? And how to guarantee semantic coherence among different token positions?**
>
> We appreciate you raising these points! In practice, we keep increasing $ k $ until $ \text{candidates}_i $ contains a minimum number of intersections. Additionally, as outlined in Eq. 3, the top-$ k $ candidates are generated by the LLM while conditioning on the original question and preceding prompt tokens. This ensures semantic coherence among the tokens, as it is inherently guided by the LLM's understanding of the context.
>
> ---
>
> > ### **Question 2**
> > **Can you explain why the performance gain from gradient over reasoning is much smaller on Gemma model compared with LLaMA model?**
>
> We agree that the performance gain from gradient over reasoning is much larger in LLaMA-3 compared to Gemma-2 models. Generally, Gemma-2-9B is a stronger model compared to LLaMA-3-8B. Therefore, we hypothesize that it already generates strong prompt candidates in the prompt token candidate proposal stage (Section 4.2). Consequently, the candidate selection stage where gradients are leveraged has less burden to find the best candidate. Nevertheless, gradient over reasoning still chooses a better token candidate, leading to improved performance.
>
> ---
>
> > ### **Question 3**
> > **How about the performance on 1.5B or 3B models? A 9B model is actually not very small in practice.**
>
> Thank you so much for raising this issue as well. While 8B and 9B models are popular, we agree that smaller models like LLaMA-3.2-1B-Instruct would be a great candidate for a very small language model. In the revised manuscript, **Table 8** shows the performance comparison of GreaTer to that of other prompts optimized by other baselines. The performance numbers clearly show that, on average, GreaTer substantially outperforms other approaches in different tasks. For your reference, the performance comparison in the LLaMA-3.2-1B target model is given below:
>
> | **Method (optimized by)**  | **movie_rec.** | **causal_judgement** | **hyperbaton** | **tracking_five** | **object_count.** | **AVG**   |
> |----------------------------|----------------|----------------------|----------------|-------------------|-------------------|----------|
> | APO (GPT4)                | 23             | 45                   | 61             | 24                | 51                | 40.8     |
> | PE2 (GPT4)                | 27             | 54                   | **69**             | 21                | 59                | 46.0     |
> | iAPE (GPT4)               | 32             | 57                   | 53             | 15                | 41                | 39.6     |
> | **GreaTer (Llama-3.2-1B)**| **46**         | **69**               | 62         | **24**            | **67**            | **53.6** |
>
> ---
>
> ### References
>
> [1] Kojima, Takeshi, et al. "Large language models are zero-shot reasoners." *Advances in Neural Information Processing Systems* 35 (2022): 22199-22213.
> [2] Shin, Taylor, et al. "Autoprompt: Eliciting knowledge from language models with automatically generated prompts." *arXiv preprint arXiv:2010.15980* (2020).
> [3] Wen, Yuxin, et al. "Hard prompts made easy: Gradient-based discrete optimization for prompt tuning and discovery." *Advances in Neural Information Processing Systems* 36 (2024).
> [4] Pryzant, Reid, et al. "Automatic prompt optimization with" gradient descent" and beam search." arXiv preprint arXiv:2305.03495 (2023).
> [5] Hu, Wenyang, et al. "Localized zeroth-order prompt optimization." arXiv preprint arXiv:2403.02993 (2024).

---

> ### Author Response · Authors · 2024-12-01
> **Kind Reminder from Authors**
>
> Dear Reviewer 7nfg,
>
> We sincerely appreciate your valuable feedback. With the discussion concluding soon (December 2nd), we kindly ask if our responses have addressed your concerns and let us know if any further questions or comments you might have. Thanks!
>
> Best,
> Authors

---

> > ### Comment · Reviewer_7nfg · 2024-12-01
> >
> > Thank you for the additional experiments and clarifications. Things are clearer now in the revised paper, and my concerns have been addressed. I am willing to increase my score.

---

### Official Review · Reviewer_ZouY · 2024-11-04

**Soundness:** 3
**Presentation:** 3
**Contribution:** 2
**Rating:** 6
**Confidence:** 3

**Summary:**

The paper introduces a prompt optimization technique that leverages gradient information over task-specific reasoning to enhance the performance of smaller language models (LLMs). GREATER enables self-optimization of prompts without relying on large, computationally expensive LLMs, which is a significant departure from existing methods that depend on textual feedback from large models.

**Strengths:**

1. GREATER presents an approach to prompt optimization by incorporating gradient information directly into the process, which is a departure from traditional text-based feedback methods. This innovation could reduce the computational costs associated with prompt engineering.

2. The paper is well-structured, with a clear problem statement and a detailed explanation of the GREATER method.

**Weaknesses:**

1. While GREATER shows promising results, it is not clear how well these findings generalize to other types of tasks beyond reasoning tasks. Future work could explore the applicability of GREATER to a broader range of NLP tasks.

2. Although GREATER aims to reduce reliance on large LLMs, the paper does not discuss the computational resources required for the gradient-based optimization process itself. More details on this aspect would be beneficial.

3. The paper could benefit from a more detailed comparison with other gradient-based methods, especially those that also aim to optimize prompts without relying on large LLMs.

**Questions:**

1. Can the authors comment on the potential of GREATER to be applied to tasks outside of reasoning, such as generation or summarization tasks?

2. How do the computational costs of GREATER compare to traditional text-based prompt optimization methods, especially when considering the training and inference phases?

3. The paper mentions the impact of different initialization prompts. Could the authors elaborate on how sensitive GREATER is to the choice of initialization and whether there are best practices for initializing prompts?

---

> ### Author Response · Authors · 2024-11-25
> **Author Response to Reviewer ZouY (Part 1/2)**
>
> Thank you so much for your insightful reviews! We have uploaded an updated PDF by including new additional experiments and improved paper presentation accordingly. We carefully address the weaknesses and questions as follows:
>
> ---
>
> >**Weakness 1**  While GreaTer shows promising results, it is not clear how well these findings generalize to other types of tasks beyond reasoning tasks. Future work could explore the applicability of GreaTer to a broader range of NLP tasks.**
>
> Thank you for bringing up this valuable point. We acknowledge that GreaTer is currently designed with a primary focus on reasoning tasks to enhance performance in complex scenarios. A critical aspect of GreaTer is the computation of gradients based on task loss, which requires well-defined and objective evaluation criteria. However, tasks like generation or summarization might not have clearly defined objective evaluation criteria.
>
> An interesting direction for future work could involve leveraging model-based verifiers to provide a reward signal for such tasks. This reward signal could then be used to compute policy gradients, enabling GreaTer to optimize prompts even for tasks without explicit evaluation criteria. Exploring this approach could broaden the applicability of GreaTer to generation, summarization, and other NLP tasks beyond reasoning.
>
> ---
>
> > **Weakness 2**  Although GreaTer aims to reduce reliance on large LLMs, the paper does not discuss the computational resources required for the gradient-based optimization process itself. More details on this aspect would be beneficial.
>
> Thank you for your feedback. We have added a detailed complexity analysis in Appendix D of the updated manuscript, comparing the computational demands of GreaTer with text-feedback-based approaches. This provides valuable insights into the resource requirements of our method. A summary of the analysis is given below:
>
> #### **Complexity Analysis: GreaTer vs. Text-Based Feedback Approaches**
> Assume each sample contains $L$ input-output tokens on average.
>
> ---
>
> ### **GreaTer**
> - **Per Sample**:
>   - Forward pass: $O(L^2)$
>   - Backward pass: $O(L^2) $
> - **Total Complexity**:
>   $O(NL^2)$
>
> ---
>
> ### **Text-Based Feedback Approaches**
> - **Per Iteration**:
>   - Forward pass for $N$ samples: $O(NL^2)$
>   - Feedback generation for $n$ incorrect samples: $O(n^2L^2) $
> - **Total Complexity**:  $O(NL^2) + O(n^2L^2)$
>
> ---
>
> ### **Comparison**
> - **GreaTer**: $ O(NL^2) $ (consistent).
> - **Text-Based Feedback**:
>   - Simple tasks ($ n \to 0 $): $ O(NL^2) $
>   - Difficult tasks ($ n \to N $): $ O(N^2L^2) $
>
>
> This complexity difference is also translated into real-world performance. In the *movie_recommendation* task, GreaTer required approximately ~5 hours for optimization, whereas TextGrad required a total of ~14 hours for prompt optimization in our own setup.
>
> ---
>
> > **Weakness 3**  The paper could benefit from a more detailed comparison with other gradient-based methods, especially those that also aim to optimize prompts without relying on large LLMs.
>
> Thank you very much for providing this suggestion! We have compared GreaTer performance to that of some gradient-based approaches in Appendix C (Table 6) of the updated manuscript. Here we compare the performance of GreaTer to that of AutoPrompt [1] and PEZ [2]. The table is summarized below:
>
> | Method       | Llama-3-8B (movie_rec.) | Llama-3-8B (tracking_five) | Gemma-2-9B (movie_rec.) | Gemma-2-9B (tracking_five) |
> |--------------|--------------------------|----------------------------|--------------------------|----------------------------|
> | AutoPrompt   | 51                    | 38                       | 50                    | 73                       |
> | PEZ          | 42                    | 35                       | 45                    | 73                       |
> | **GreaTer**  | **56**                | **70**                   | **57**                | **86**                   |
>
> GreaTer also provides interpretable prompts, whereas baselines provide gibberish prompts (e.g., PEZ provides “ılığı, nullabancı Prince numerous-source sweeps Sawyer62337” as the optimized prompt for *movie_recommendation*). Overall, this table clearly demonstrates that GreaTer significantly outperforms previous gradient-based approaches, primarily due to its adoption of gradient over reasoning. In contrast, these baselines overlook reasoning when calculating both the task loss and gradient, resulting in suboptimal signals for prompt optimization in reasoning tasks.
>
> ---
>
> > **Question 1**  Can the authors comment on the potential of GreaTer to be applied to tasks outside of reasoning, such as generation or summarization tasks?
>
> Thanks for bringing this up! Please check our response to Weakness 1.

---

> ### Author Response · Authors · 2024-11-25
> **Author Response to Reviewer ZouY (Part 2/2)**
>
> > **Question 2**  How do the computational costs of GreaTer compare to traditional text-based prompt optimization methods, especially when considering the training and inference phases?
>
> Thanks for asking this! We include information on the computational costs of GreaTer in our response to **Weakness 2**.
>
> ---
>
> > **Question 3**  The paper mentions the impact of different initialization prompts. Could the authors elaborate on how sensitive GreaTer is to the choice of initialization and whether there are best practices for initializing prompts?
>
> Thanks for asking this! As shown in **Table 7** of Appendix F, different initialization prompts can lead to prompts that look quite distinct yet achieve comparable performance metrics. This variation arises because there are often multiple valid ways (prompts) to solve a problem, and the optimization process may converge to any of these depending on the initialization.
>
> Similar to other optimization problems, a well-chosen initialization can be beneficial. While we use a fixed initialization prompt across all problems in our experiments, incorporating human insights or domain-specific understanding into the initialization can guide the optimization process toward the target prompt more efficiently, potentially enabling faster convergence.
>
> ---
>
> ### References
> [1] Shin, Taylor, et al. "Autoprompt: Eliciting knowledge from language models with automatically generated prompts." *arXiv preprint arXiv:2010.15980* (2020).
> [2] Wen, Yuxin, et al. "Hard prompts made easy: Gradient-based discrete optimization for prompt tuning and discovery." *Advances in Neural Information Processing Systems* 36 (2024).

---

> ### Author Response · Authors · 2024-12-01
> **Kind Reminder from Authors**
>
> Dear Reviewer ZouY,
>
> We sincerely appreciate your valuable feedback. With the discussion concluding soon (December 2nd), we kindly ask if our responses have addressed your concerns and let us know if any further questions or comments you might have. Thanks!
>
> Best,
> Authors

---

### Official Review · Reviewer_24Zw · 2024-11-04

**Soundness:** 3
**Presentation:** 3
**Contribution:** 2
**Rating:** 8
**Confidence:** 3

**Summary:**

The paper presents GREATER, a novel prompt optimization technique that enables smaller language models to self-optimize their prompts without relying on larger, costly language models.

**Strengths:**

* This paper addresses a significant limitation in current prompt optimization methods
* This paper involves gradient information to prompt optimization
* This paper shows practical utility for improving smaller language model performance
* Figure 1 clearly demonstrates the difference between GREATER and previous methods

**Weaknesses:**

* The case studies section could benefit from more detailed analysis of why certain prompts work better
* Could benefit from more theoretical analysis of why gradients over reasoning work better
* Line 462 explicitly mentions large language models (LLMs) which have already been defined in the introduction section. There seems like a typo in line 1186 as well, "Use these logical reasoning process steps and explain Step. step. Here is correct answer.","step" appears multiple times.
* Could include more diverse task types beyond reasoning tasks

**Questions:**

* How does the method scale with increasing model size? Have you considered using GREATER on larger open-source models like Llama-3.1-70B?

---

> ### Author Response · Authors · 2024-11-25
> **Author Response to Reviewer 24Zw (Part 1/2)**
>
> Thank you so much for your thoughtful and constructive feedback! Your insights and suggestions have been extremely valuable in improving our manuscript. We have uploaded an updated PDF by including new additional experiments and improved paper presentation accordingly. Below, we have carefully addressed each of the points you raised.
>
> ---
>
> > **Weakness 1:** The case studies section could benefit from more detailed analysis of why certain prompts work better.
>
> Thank you for your insightful suggestion on this point. Well-designed prompts do more than give basic instructions—they provide clear, practical strategies tailored to the task. By encouraging structured and systematic thinking, such prompts improve problem-solving and enhance model performance. We can show one example from the BBH-navigate task, demonstrating how different prompts affect the responses:
>
> ---
> #### **Question**
> If you follow these instructions, do you return to the starting point?
> - Take 7 steps.
> - Take 4 steps.
> - Take 3 steps.
> - Take 10 steps.
> - Turn left.
> - Turn around.
>
> **Options:**
> - Yes
> - No
>
> ---
>
> #### **APO Prompt:** Let's think step by step.
>
> **Response:**
> Let’s think step by step.
>
> 1. **7 steps:** You move 7 steps forward.
> 2. **4 steps:** You move 4 more steps forward.
> 3. **3 steps:** You move 3 more steps forward.
> 4. **10 steps:** You move 10 more steps forward.
> 5. **Turn left:** You change direction to the left.
> 6. **Turn around:** You turn 180 degrees, now facing the opposite direction.
>
> At this point, you've essentially reversed your path. Since you turned around, you'll be heading back towards your starting point.
>
> **Answer:** Yes. **[Incorrect]**
>
> ---
>
> #### **GreaTer Prompt:** Use proper mathematical logic, explaining step by step. Finally, give the correct answer.
>
> **Response:**
> 1. **Step 1:** Take 7 steps forward.
> 2. **Step 2:** Take 4 steps forward.
> 3. **Step 3:** Take 3 steps forward.
> 4. **Step 4:** Take 10 steps forward.
>
>    At this point, you've taken a total of $7 + 4 + 3 + 10 = 24$ steps forward.
>
> 5. **Step 5:** Turn left. This changes your direction but doesn't affect the total distance traveled.
> 6. **Step 6:** Turn around. This means you're now facing the opposite direction of where you started.
> 7. **Step 7:** No further steps are taken in this instruction set.
>
> **Answer:** No.
> You will not return to the starting point because you've moved $24$ steps forward and then simply turned around. **[Correct]**
>
> ---
>
> From this example, we can see that the GreaTer prompt encouraged the LLM to solve the problem from a mathematical perspective, calculating the steps accurately, which led to the correct response. On the contrary, the APO prompt lacked such guidance, resulting in an incorrect conclusion. We have expanded this discussion in the case studies section of the draft.
>
> ---
>
> > **Weakness 2:** Could benefit from more theoretical analysis of why gradients over reasoning work better.
>
> GreaTer employs a well-defined optimization framework, as detailed in Section 4 of our paper. Our method explicitly integrates reasoning into the optimization process. The steps involve generating reasoning, calculating the loss, and then using the gradient over the reasoning to refine the prompt. This approach ensures reasoning is a core part of the optimization. For example:
>
> **Input ($x$):** "The cafeteria had 23 apples. If they used 20 to make lunch and bought 6 more, how many apples do they have?"
> **Prompt ($p$):** "Use logical reasoning and think step by step."
> **Reasoning chain ($r$):** "The cafeteria had 23 apples originally. They used 20 to make lunch. So they had $23 - 20 = 3$. They bought 6 more, so they now have $3 + 6 = 9$."
>
> When conditioning the model on $x \odot p$ (input and prompt only), the reasoning is excluded, often resulting in noisy gradients and misaligned probabilities. The model processes:
>
> “The cafeteria had 23 apples. ... how many apples do they have? Use logical reasoning and think step by step. The final answer is …”
>
> Here, the next token probabilities are not conditioned on the reasoning chain, leading to incorrect logits and suboptimal loss values.
>
> In contrast, conditioning on $x \odot p \odot r$ (input, prompt, and reasoning) explicitly incorporates the reasoning chain. The model processes:
>
> “The cafeteria had 23 apples. ... how many apples do they have? Use logical reasoning and think step by step ... The cafeteria had 23 apples originally. They used 20 to make lunch. … $3 + 6 = 9$. Therefore, the final answer is …”
>
> In this case, aligning token probabilities with the reasoning chain improves logits and minimizes loss. By explicitly integrating reasoning, the model enhances both reasoning and task performance, explaining why GreaTer outperforms methods that omit this integration.

---

> ### Author Response · Authors · 2024-11-25
> **Author Response to Reviewer 24Zw (Part 2/2)**
>
> ---
>
> > **Weakness 3:** Line 462 explicitly mentions large language models (LLMs), which have already been defined in the introduction section. There seems to be a typo in Line 1186 as well: "Use these logical reasoning process steps and explain Step. step. Here is correct answer.","step" appears multiple times.
>
> Thank you for catching these issues! We have addressed the typo on Line 462 in the revised manuscript. Regarding Line 1186, the text represents an optimized prompt and is an example where minor grammatical inconsistencies are present. However, we believe these prompts still remain interpretable and significantly better compared to the unintelligible prompts generated by some other methods. For example, in our comparison with the prior gradient-based approach, PEZ, one of its optimized prompts was *"ıldığı, nullabancı Prince numerous-source sweeps Sawyer62337"*
>
> As evident, such prompts are largely gibberish and lack interpretability, underscoring the advantage of GreaTer in generating more coherent and functional prompts.
>
> ---
>
> > **Weakness 4:** Could include more diverse task types beyond reasoning tasks.
>
> Thank you for highlighting this point. GreaTer is currently focused on reasoning tasks, leveraging gradients computed from well-defined task loss for optimization. Tasks like generation, however, often lack clear objective evaluation criteria. A promising future direction could involve using model-based verifiers to provide reward signals for such tasks. This can extend GreaTer’s applicability to diverse NLP tasks (including generation/summarization tasks) by using LLM verifier-provided reward signals to compute policy gradients for token selection.
>
> ---
>
> > **Question 1:** How does the method scale with increasing model size? Have you considered using GreaTer on larger open-source models like Llama-3.1-70B?
>
> We really appreciate you bringing up this point. While GreaTer is primarily focused on smaller language models, it would definitely be very interesting to use it on larger models like Llama-3.1-70B. By guiding these powerful models with optimized prompts, we could potentially tackle significantly more challenging tasks. However, scaling to such a large model will require more computational resources and is currently beyond our available computational constraints.
>
> Since our primary focus is on smaller language models, we added a new experiment by comparing the performance of GreaTer in a very small language model Llama-3.2-1B to the baselines. The table is summarized below. As we can see, for very small language models, GreaTer provides excellent performance across different tasks.
>
> ---
>
> | **Method (optimized by)**  | **movie_rec.** | **causal_judgement** | **hyperbaton** | **tracking_five** | **object_count.** | **AVG**   |
> |-----------------------------|----------------|----------------------|----------------|-------------------|-------------------|-----------|
> | APO (GPT4)                 | 23             | 45                   | 61             | 24                | 51                | 40.8      |
> | PE2 (GPT4)                 | 27             | 54                   | **69**         | 21                | 59                | 46.0      |
> | iAPE (GPT4)                | 32             | 57                   | 53             | 15                | 41                | 39.6      |
> | GreaTer (Llama-3.2-1B)     | **46**         | **69**               | 62             | **24**            | **67**            | **53.6**  |
>
> ---
> We hope these changes and additions address all your concerns and provide further clarity to our approach!

---

> ### Author Response · Authors · 2024-12-01
> **Kind Reminder from Authors**
>
> Dear Reviewer 24Zw,
>
> We sincerely appreciate your valuable feedback. With the discussion concluding soon (December 2nd), we kindly ask if our responses have addressed your concerns and let us know if any further questions or comments you might have. Thanks!
>
> Best,
> Authors

---

### Meta-Review · Area_Chair_rx9X · 2024-12-19

**Metareview:**

This paper introduces a gradient-based approach to optimize a set of candidate prompts towards a target task, represented by a data sample that can be used for "learning" the prompts. The proposal is akin to recent methods that leverage an external model to iteratively improve the prompt, though as authors argue, requiring access to that external model is a major limitation of existing methods, and the proposal in the paper, dubbed GReaTer, offers a self-improving alternative. The approach works by performing gradient descent steps over one-hot vectors that continuously represent the vocabulary elements. In each time-step, continuous gradient updates are performed on one-hot codes of the inputs, and the tokens are replaced by the argmax of the one-hot representation after the update.

The reviewers are unanimously in favor of acceptance, and I side with them since the community could clearly benefit of more efficient prompt tuning frameworks. I would nonetheless note limitations of the approach in that it requires full access to the model one wants to optimize for, while tuning methods that rely on external auxiliary prompt optimizers also support black box language models. I would also add that, as highlighted by the authors, the proposal could yield prompts that are not grammatically correct, or even more so, non-natural (or unsafe/unethical) prompts that still perform well in the task at hand could be obtained. An investigation on to what extent that risk is relevant or at least an in-depth discussion is missing from the paper.

**Additional Comments On Reviewer Discussion:**

One important concern highlighted by some of the reviewers is the cost in time to optimize a prompt. Authors did reply satisfactorily however with a complexity table and performance data.

---

### Decision · Program_Chairs · 2025-01-22

Accept (Poster)